# Higher-Order Total Variation Classes on Grids: Minimax Theory and Trend Filtering Methods

**Veeranjaneyulu Sadhanala**
Carnegie Mellon University
Pittsburgh, PA 15213
vsadhana@cs.cmu.edu

**Yu-Xiang Wang**
Carnegie Mellon University/Amazon AI
Pittsburgh, PA 15213/Palo Alto, CA 94303
yuxiangw@amazon.com

**James Sharpnack**
University of California, Davis
Davis, CA 95616
jsharpna@ucdavis.edu

**Ryan J. Tibshirani**
Carnegie Mellon University
Pittsburgh, PA 15213
ryantibs@stat.cmu.edu

## Abstract

We consider the problem of estimating the values of a function over $n$ nodes of a $d$-dimensional grid graph (having equal side lengths $n^{1/d}$) from noisy observations. The function is assumed to be smooth, but is allowed to exhibit different amounts of smoothness at different regions in the grid. Such heterogeneity eludes classical measures of smoothness from nonparametric statistics, such as Holder smoothness. Meanwhile, total variation (TV) smoothness classes allow for heterogeneity, but are restrictive in another sense: only constant functions count as perfectly smooth (achieve zero TV). To move past this, we define two new higher-order TV classes, based on two ways of compiling the discrete derivatives of a parameter across the nodes. We relate these two new classes to Holder classes, and derive lower bounds on their minimax errors. We also analyze two naturally associated trend filtering methods; when $d = 2$, each is seen to be rate optimal over the appropriate class.

## 1 Introduction

In this work, we focus on estimation of a mean parameter defined over the nodes of a $d$-dimensional grid graph $G = (V, E)$, with equal side lengths $N = n^{1/d}$. Let us enumerate $V = \{1, \ldots, n\}$ and $E = \{e_1, \ldots, e_m\}$, and consider data $y = (y_1, \ldots, y_n) \in \mathbb{R}^n$ observed over $V$, distributed as

$$y_i \sim N(\theta_{0,i}, \sigma^2), \quad \text{independently, for } i = 1, \ldots, n, \tag{1}$$

where $\theta_0 = (\theta_{0,1}, \ldots, \theta_{0,n}) \in \mathbb{R}^n$ is the mean parameter to be estimated, and $\sigma^2 > 0$ the common noise variance. We will assume that $\theta_0$ displays some kind of regularity or smoothness over $G$, and are specifically interested in notions of regularity built around on the *total variation (TV)* operator

$$\|D\theta\|_1 = \sum_{(i,j) \in E} |\theta_i - \theta_j|, \tag{2}$$

defined with respect to $G$, where $D \in \mathbb{R}^{m \times n}$ is the edge incidence matrix of $G$, which has $\ell$th row $D_\ell = (0, \ldots, -1, \ldots, 1, \ldots, 0)$, with $-1$ in location $i$ and $1$ in location $j$, provided that the $\ell$th edge is $e_\ell = (i, j)$ with $i < j$. There is an extensive literature on estimators based on TV regularization, both in Euclidean spaces and over graphs. Higher-order TV regularization, which, loosely speaking, considers the TV of derivatives of the parameter, is much less understood, especially over graphs. In this paper, we develop statistical theory for higher-order TV smoothness classes, and we analyze

associated trend filtering methods, which are seen to achieve the minimax optimal estimation error rate over such classes. This can be viewed as an extension of the work in [22] for the zeroth-order TV case, where by "zeroth-order", we refer to the usual TV operator as defined in (2).

**Motivation.** TV denoising over grid graphs, specifically 1d and 2d grid graphs, is a well-studied problem in signal processing, statistics, and machine learning, some key references being [20, 5, 26]. Given data $y \in \mathbb{R}^n$ as per the setup described above, the *TV denoising* or *fused lasso* estimator over the grid $G$ is defined as

$$\hat{\theta} = \underset{\theta \in \mathbb{R}^n}{\operatorname{argmin}} \ \frac{1}{2} \|y - \theta\|_2^2 + \lambda \|D\theta\|_1, \tag{3}$$

where $\lambda \geq 0$ is a tuning parameter. The TV denoising estimator generalizes seamlessly to arbitrary graphs. The problem of denoising over grids, the setting we focus on, is of particular relevance to a number of important applications, e.g., in time series analysis, and image and video processing.

A strength of the nonlinear TV denoising estimator in (3)—where by "nonlinear", we mean that $\hat{\theta}$ is nonlinear as a function of $y$—is that it can adapt to heterogeneity in the local level of smoothness of the underlying signal $\theta_0$. Moreover, it adapts to such heterogeneity at an extent that is beyond what linear estimators are capable of capturing. This principle is widely evident in practice and has been championed by many authors in the signal processing literature. It is also backed by statistical theory, i.e., [8, 16, 27] in the 1d setting, and most recently [22] in the general $d$-dimensional setting.

Note that the TV denoising estimator $\hat{\theta}$ in (3) takes a *piecewise constant* structure by design, i.e., at many adjacent pairs $(i, j) \in E$ we will have $\hat{\theta}_i = \hat{\theta}_j$, and this will be generally more common for larger $\lambda$. For some problems, this structure may not be ideal and we might instead seek a *piecewise smooth* estimator, that is still able to cope with local changes in the underlying level of smoothness, but offers a richer structure (beyond a simple constant structure) for the base trend. In a 1d setting, this is accomplished by trend filtering methods, which move from piecewise constant to *piecewise polynomial* structure, via TV regularization of discrete derivatives of the parameter [24, 13, 27]. An extension of trend filtering to general graphs was developed in [31]. In what follows, we study the statistical properties of this graph trend filtering method over grids, and we propose and analyze a more specialized trend filtering estimator for grids based on the idea that something like a Euclidean coordinate system is available at any (interior) node. See Figure 1 for a motivating illustration.

**Related work.** The literature on TV denoising is enormous and we cannot give a comprehensive review, but only some brief highlights. Important methodological and computational contributions are found in [20, 5, 26, 4, 10, 6, 28, 15, 7, 12, 1, 25], and notable theoretical contributions are found in [16, 19, 9, 23, 11, 22, 17]. The literature on higher-order TV-based methods is more sparse and more concentrated on the 1d setting. Trend filtering methods in 1d were pioneered in [24, 13], and analyzed statistically in [27], where they were also shown to be asymptotically equivalent to locally adaptive regression splines of [16]. An extension of trend filtering to additive models was given in [21]. A generalization of trend filtering that operates over an arbitrary graph structure was given in [31]. Trend filtering is not the only avenue for higher-order TV regularization: the signal processing community has also studied higher-order variants of TV, see, e.g., [18, 3]. The construction of the discrete versions of these higher-order TV operators is somewhat similar to that in [31] as well our Kronecker trend filtering proposal, however, the focus of the work is quite different.

**Summary of contributions.** An overview of our contributions is given below.

- We propose a new method for trend filtering over grid graphs that we call *Kronecker trend filtering* (KTF), and compare its properties to the more general graph trend filtering (GTF) proposal of [31].

- For 2d grids, we derive estimation error rates for GTF and KTF, each of these rates being a function of the regularizer evaluated at the mean $\theta_0$.

- For $d$-dimensional grids, we derive minimax lower bounds for estimation over two higher-order TV classes defined using the operators from GTF and KTF. When $d = 2$, these lower bounds match the upper bounds in rate (apart from log factors) derived for GTF and KTF, ensuring that each method is minimax rate optimal (modulo log factors) for its own notion of regularity. Also, the KTF class contains a Holder class of an appropriate order, and KTF is seen to be rate optimal (modulo log factors) for this more homogeneous class as well.

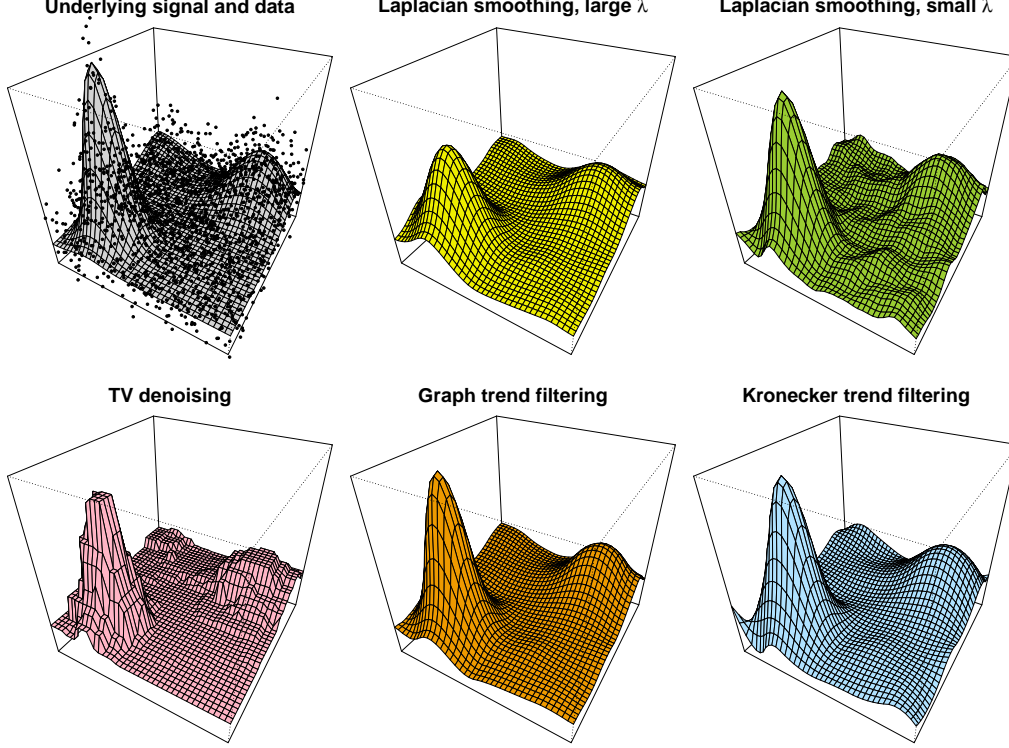

Figure 1: *Top left: an underlying signal $\theta_0$ and associated data $y$ (shown as black points). Top middle and top right: Laplacian smoothing fit to $y$, at large and small tuning parameter values, respectively. Bottom left, middle, and right: TV denoising (3), graph trend filtering (5), and Kronecker trend filtering (5) fit to $y$, respectively (the latter two are of order $k = 2$, with penalty operators as described in Section 2). In order to capture the larger of the two peaks, Laplacian smoothing must significantly undersmooth throughout; with more regularization, it undersmooths throughout. TV denoising is able to adapt to heterogeneity in the smoothness of the underlying signal, but exhibits "staircasing" artifacts, as it is restricted to fitting piecewise constant functions. Graph and Kronecker trend filtering overcome this, while maintaining local adaptivity.*

**Notation.** For deterministic sequences $a_n, b_n$ we write $a_n = O(b_n)$ to denote that $a_n/b_n$ is upper bounded for large enough $n$, and $a_n \asymp b_n$ to denote that both $a_n = O(b_n)$ and $a_n^{-1} = O(b_n^{-1})$. For random sequences $A_n, B_n$, we write $A_n = O_{\mathbb{P}}(B_n)$ to denote that $A_n/B_n$ is bounded in probability.

Given a $d$-dimensional grid $G = (V, E)$, where $V = \{1, \ldots, n\}$, as before, we will sometimes index a parameter $\theta \in \mathbb{R}^n$ defined over the nodes in the following convenient way. Letting $N = n^{1/d}$ and $Z_d = \{(i_1/N, \ldots, i_d/N) : i_1, \ldots, i_d \in \{1, \ldots, N\}\} \subseteq [0,1]^d$, we will index the components of $\theta$ by their lattice positions, denoted $\theta(x)$, $x \in Z_d$. Further, for each $j = 1, \ldots, d$, we will define the discrete derivative of $\theta$ in the $j$th coordinate direction at a location $x$ by

$$(D_{x_j}\theta)(x) = \begin{cases} \theta(x + e_j/N) - \theta(x) & \text{if } x, x + e_j/N \in Z_d, \\ 0 & \text{else.} \end{cases} \tag{4}$$

Naturally, we denote by $D_{x_j}\theta \in \mathbb{R}^n$ the vector with components $(D_{x_j}\theta)(x)$, $x \in Z_d$. Higher-order discrete derivatives are simply defined by repeated application of the above definition. We use abbreviations $(D_{x_j^2}\theta)(x) = (D_{x_j}(D_{x_j}\theta))(x)$, for $j = 1, \ldots, d$, and $(D_{x_j,x_\ell}\theta)(x) = (D_{x_j}(D_{x_\ell}\theta))(x)$, for $j, \ell = 1, \ldots, d$, and so on.

Given an estimator $\hat{\theta}$ of the mean parameter $\theta_0$ in (1), and $\mathcal{K} \subseteq \mathbb{R}^n$, two quantities of interest are:

$$\text{MSE}(\hat{\theta}, \theta_0) = \frac{1}{n}\|\hat{\theta} - \theta_0\|_2^2 \quad \text{and} \quad R(\mathcal{K}) = \inf_{\hat{\theta}} \sup_{\theta_0 \in \mathcal{K}} \mathbb{E}\big[\text{MSE}(\hat{\theta}, \theta_0)\big].$$

The first quantity here is called the mean squared error (MSE) of $\theta$; we will also call $\mathbb{E}[\text{MSE}(\hat{\theta}, \theta_0)]$ the risk of $\hat{\theta}$. The second quantity is called the minimax risk over $\mathcal{K}$ (the infimum being taken over all estimators $\hat{\theta}$).

## 2   Trend filtering methods

**Review: graph trend filtering.**   To review the family of estimators developed in [31], we start by introducing a general-form estimator called the *generalized lasso* signal approximator [28],

$$\hat{\theta} = \underset{\theta \in \mathbb{R}^n}{\operatorname{argmin}} \ \frac{1}{2} \|y - \theta\|_2^2 + \lambda \|\Delta\theta\|_1, \tag{5}$$

for a matrix $\Delta \in \mathbb{R}^{r \times n}$, referred to as the penalty operator. For an integer $k \geq 0$, the authors [31] defined the *graph trend filtering* (GTF) estimator of order $k$ by (5), with the penalty operator being

$$\Delta^{(k+1)} = \begin{cases} DL^{k/2} & \text{for } k \text{ even}, \\ L^{(k+1)/2} & \text{for } k \text{ odd}. \end{cases} \tag{6}$$

Here, as before, we use $D$ for the edge incidence matrix of $G$. We also use $L = D^T D$ for the graph Laplacian matrix of $G$. The intuition behind the above definition is that $\Delta^{(k+1)}\theta$ gives something roughly like the $(k+1)$st order discrete derivatives of $\theta$ over the graph $G$.

Note that the GTF estimator reduces to TV denoising in (3) when $k = 0$. Also, like TV denoising, GTF applies to arbitrary graph structures; see [31] for more details and for the study of GTF over general graphs. Our interest is of course its behavior over grids, and we will now use the notation introduced in (4), to shed more light on the GTF penalty operator in (6) over a $d$-dimensional grid. For any signal $\theta \in \mathbb{R}^n$, we can write $\|\Delta^{(k+1)}\theta\|_1 = \sum_{x \in Z_d} d_x$, where at all points $x \in Z_d$ (except for those close to the boundary),

$$d_x = \begin{cases} \sum_{j_1=1}^d \left| \sum_{j_2,\ldots,j_q=1}^d \left( D_{x_{j_1}, x_{j_2}^2, \ldots, x_{j_q}^2} \theta \right)(x) \right| & \text{for } k \text{ even, where } q = k/2, \\ \left| \sum_{j_1,\ldots,j_q=1}^d \left( D_{x_{j_1}^2, x_{j_2}^2, \ldots, x_{j_q}^2} \theta \right)(x) \right| & \text{for } k \text{ odd, where } q = (k+1)/2. \end{cases} \tag{7}$$

Written in this form, it appears that the GTF operator $\Delta^{(k+1)}$ aggregates derivatives in somewhat of an unnatural way. But we must remember that for a general graph structure, only first derivatives and divergences have obvious discrete analogs—given by application of $D$ and $L$, respectively. Hence, GTF, which was originally designed for general graphs, relies on combinations of $D$ and $L$ to produce something like higher-order discrete derivatives. This explains the form of the aggregated derivatives in (6), which is entirely based on divergences.

**Kronecker trend filtering.**   There is a natural alternative to the GTF penalty operator that takes advantage of the Euclidean-like structure available at the (interior) nodes of a grid graph. At a point $x \in Z_d$ (not close to the boundary), consider using

$$d_x = \sum_{j=1}^d \left| \left( D_{x_j^{k+1}} \theta \right)(x) \right| \tag{8}$$

as a basic building block for penalizing derivatives, rather than (7). This gives rise to a method we call *Kronecker trend filtering* (KTF), which for an integer order $k \geq 0$ is defined by (5), but now with the choice of penalty operator

$$\widetilde{\Delta}^{(k+1)} = \begin{bmatrix} D_{1d}^{(k+1)} \otimes I \otimes \cdots \otimes I \\ I \otimes D_{1d}^{(k+1)} \otimes \cdots \otimes I \\ \vdots \\ I \otimes I \otimes \cdots \otimes D_{1d}^{(k+1)} \end{bmatrix}. \tag{9}$$

Here, $D_{1d}^{(k+1)} \in \mathbb{R}^{(N-k-1) \times N}$ is the 1d discrete derivative operator of order $k+1$ (e.g., as used in univariate trend filtering, see [27]), $I \in \mathbb{R}^{N \times N}$ is the identity matrix, and $A \otimes B$ is the Kronecker product of matrices $A, B$. Each group of rows in (9) features a total of $d-1$ Kronecker products.

KTF reduces to TV denoising in (3) when $k = 0$, and thus also to GTF with $k = 0$. But for $k \geq 1$, GTF and KTF are different estimators. A look at the action of their penalty operators, as displayed in

(7), (8) reveals some of their differences. For example, we see that GTF considers mixed derivatives of total order $k + 1$, but KTF only considers directional derivatives of order $k + 1$ that are parallel to the coordinate axes. Also, GTF penalizes aggregate derivatives (i.e., sums of derivatives), whereas KTF penalizes individual ones.

More subtle differences between GTF and KTF have to do with the structure of their estimates, as we discuss next. Another subtle difference lies in how the GTF and KTF operators (6), (9) relate to more classical notions of smoothness, particularly, Holder smoothness. This is covered in Section 4.

**Structure of estimates.** It is straightforward to see that the GTF operator (6) has a 1-dimensional null space, spanned by $\mathbb{1} = (1, \ldots, 1) \in \mathbb{R}^n$. This means that GTF lets constant signals pass through unpenalized, but nothing else; or, in other words, it preserves the projection of $y$ onto the space of constant signals, $\bar{y}\mathbb{1}$, but nothing else. The KTF operator, meanwhile, has a much richer null space.

**Lemma 1.** *The null space of the KTF operator* (9) *has dimension* $(k + 1)^d$*, and it is spanned by a polynomial basis made up of elements*

$$p(x) = x_1^{a_1} x_2^{a_2} \cdots x_d^{a_d}, \quad x \in Z_d,$$

*where* $a_1, \ldots, a_d \in \{0, \ldots, k\}$.

The proof is elementary and (as with all proofs in this paper) is given in the supplement. The lemma shows that KTF preserves the projection of $y$ onto the space of polynomials of max degree $k$, i.e., lets much more than just constant signals pass through unpenalized.

Beyond the differences in these base trends (represented by their null spaces), GTF and KTF admit estimates with similar but generally different structures. KTF has the advantage that this structure is more transparent: its estimates are piecewise polynomial functions of max degree $k$, with generally fewer pieces for larger $\lambda$. This is demonstrated by a functional representation for KTF, given next.

**Lemma 2.** *Let* $h_i : [0, 1] \to \mathbb{R}$, $i = 1, \ldots, N$ *be the (univariate) falling factorial functions [27, 30] of order* $k$*, defined over knots* $1/N, 2/N, \ldots, N$:

$$h_i(t) = \prod_{\ell=1}^{i-1} (t - t^\ell), \quad t \in [0, 1], \ i = 1, \ldots, k + 1,$$

$$h_{i+k+1}(t) = \prod_{\ell=1}^{k} \left( t - \frac{i+\ell}{N} \right) \cdot \mathbb{1}\left\{ t > \frac{i+k}{N} \right\}, \quad t \in [0, 1], \ i = 1, \ldots, N - k - 1. \tag{10}$$

*(For* $k = 0$*, our convention is for the empty product to equal 1.) Let* $H_d$ *be the space spanned by all* $d$*-wise tensor products of falling factorial functions, i.e.,* $H_d$ *contains* $f : [0, 1]^d \to \mathbb{R}$ *of the form*

$$f(x) = \sum_{i_1, \ldots, i_d = 1}^{N} \alpha_{i_1, \ldots, i_d} h_{i_1}(x) h_{i_2}(x_2) \cdots h_{i_d}(x_d), \quad x \in [0, 1]^d,$$

*for coefficients* $\alpha \in \mathbb{R}^n$ *(whose components we index by* $\alpha_{i_1, \ldots, i_d}$*, for* $i_1, \ldots, i_d = 1, \ldots, N$*). Then the KTF estimator defined in* (5), (9) *is equivalent to the functional optimization problem*

$$\hat{f} = \underset{f \in H_d}{\operatorname{argmin}} \ \frac{1}{2} \sum_{x \in Z_d} \left( y(x) - f(x) \right)^2 + \lambda \sum_{j=1}^{d} \sum_{x_{-j} \in Z_{d-1}} \mathrm{TV}\left( \frac{\partial^k f(\cdot, x_{-j})}{\partial x_j^k} \right), \tag{11}$$

*where* $f(\cdot, x_{-j})$ *denotes* $f$ *as function of the* $j$*th dimension with all other dimensions fixed at* $x_{-j}$*,* $\partial^k / \partial x_j^k (\cdot)$ *denotes the* $k$*th partial weak derivative operator with respect to* $x_j$*, for* $j = 1, \ldots, d$*, and* $\mathrm{TV}(\cdot)$ *denotes the total variation operator. The discrete* (5), (9) *and functional* (11) *representations are equivalent in that* $\hat{f}$ *and* $\hat{\theta}$ *match at all grid locations* $x \in Z_d$.

Aside from shedding light on the structure of KTF solutions, the functional optimization problem in (11) is of practical importance: the function $\hat{f}$ is defined over all of $[0, 1]^d$ (as opposed to $\hat{\theta}$, which is of course only defined on the grid $Z_d$) and thus we may use it to interpolate the KTF estimate to non-grid locations. It is not clear to us that a functional representation as in (11) (or even a sensible interpolation strategy) is available for GTF on $d$-dimensional grids.

# 3 Upper bounds on estimation error

In this section, we assume that $d = 2$, and derive upper bounds on the estimation error of GTF and KTF for 2d grids. Upper bounds for generalized lasso estimators were studied in [31], and we will leverage one of their key results, which is based on what these authors call incoherence of the left singular vectors of the penalty operator $\Delta$. A slightly refined version of this result is stated below.

**Theorem 1** (Theorem 6 in [31]). *Suppose that $\Delta \in \mathbb{R}^{r \times n}$ has rank $q$, and denote by $\xi_1 \leq \ldots \leq \xi_q$ its nonzero singular values. Also let $u_1, \ldots, u_q$ be the corresponding left singular vectors. Assume that these vectors, except for the first $i_0$, are incoherent, meaning that for a constant $\mu \geq 1$,*

$$\|u_i\|_\infty \leq \mu/\sqrt{n}, \quad i = i_0 + 1, \ldots, q,$$

*Then for $\lambda \asymp \mu \sqrt{(\log r/n) \sum_{i=i_0+1}^{q} \xi_i^{-2}}$, the generalized lasso estimator $\hat{\theta}$ in (5) satisfies*

$$\mathrm{MSE}(\hat{\theta}, \theta_0) = O_{\mathbb{P}}\left( \frac{\mathrm{nullity}(\Delta)}{n} + \frac{i_0}{n} + \frac{\mu}{n} \sqrt{\frac{\log r}{n} \sum_{i=i_0+1}^{q} \frac{1}{\xi_i^2}} \cdot \|\Delta\theta_0\|_1 \right).$$

For GTF and KTF, we will apply this result, balancing an appropriate choice of $i_0$ with the partial sum of reciprocal squared singular values $\sum_{i=i_0+1}^{q} \xi_i^{-2}$. The main challenge, as we will see, is in establishing incoherence of the singular vectors.

**Error bounds for graph trend filtering.** The authors in [31] have already used Theorem 1 (their Theorem 6) in order to derive error rates for GTF on 2d grids. However, their results (specifically, their Corollary 8) can be refined using a tighter upper bound for the partial sum term $\sum_{i=i_0+1}^{q} \xi_i^{-2}$. No real further tightening is possible, since, as we show later, the results below match the minimax lower bound in rate, up to log factors.

**Theorem 2.** *Assume that $d = 2$. For $k = 0$, $C_n = \|\Delta^{(1)}\theta_0\|_1$ (i.e., $C_n$ equal to the TV of $\theta_0$, as in (2)), and $\lambda \asymp \log n$, the GTF estimator in (5), (6) (i.e., the TV denoising estimator in (3)) satisfies*

$$\mathrm{MSE}(\hat{\theta}, \theta_0) = O_{\mathbb{P}}\left( \frac{1}{n} + \frac{\log n}{n} C_n \right).$$

*For any integer $k \geq 1$, $C_n = \|\Delta^{(k+1)}\theta_0\|_1$ and $\lambda \asymp n^{\frac{k}{k+2}}(\log n)^{\frac{1}{k+2}} C_n^{-\frac{k}{k+2}}$, GTF satisfies*

$$\mathrm{MSE}(\hat{\theta}, \theta_0) = O_{\mathbb{P}}\left( \frac{1}{n} + n^{-\frac{2}{k+2}}(\log n)^{\frac{1}{k+2}} C_n^{\frac{2}{k+2}} \right).$$

**Remark 1.** The result for $k = 0$ in Theorem 2 was essentially already established by [11] (a small difference is that the above rate is sharper by a factor of $\log n$; though to be fair, [11] also take into account $\ell_0$ sparsity). It is interesting to note that the case $k = 0$ appears to be quite special, in that the GTF estimator, i.e., TV denoising estimator, is adaptive to the underlying smoothness parameter $C_n$ (the prescribed choice of tuning parameter $\lambda \asymp \log n$ does not depend on $C_n$).

The technique for upper bounding $\sum_{i=i_0+1}^{q} \xi_i^{-2}$ in the proof of Theorem 2 can be roughly explained as follows. The GTF operator $\Delta^{(k+1)}$ on a 2d grid has squared singular values:

$$\left( 4\sin^2 \frac{\pi(i_1 - 1)}{2N} + 4\sin^2 \frac{\pi(i_2 - 1)}{2N} \right)^{k+1}, \quad i_1, i_2 = 1, \ldots, N.$$

We can upper bound the sum of squared reciprocal singular values with a integral over $[0,1]^2$, make use of the identity $\sin x \geq x/2$ for small enough $x$, and then switch to polar coordinates to calculate the integral (similar to [11], in analyzing TV denoising). The arguments to verify incoherence of the left singular vectors of $\Delta^{(k+1)}$ are themselves somewhat delicate, but were already given in [31].

**Error bounds for Kronecker trend filtering.** In comparison to the GTF case, the application of Theorem 1 to KTF is a much more difficult task, because (to the best of our knowledge) the KTF operator $\tilde{\Delta}^{(k+1)}$ does not admit closed-form expressions for its singular values and vectors. This is true in any dimension (i.e., even for $d = 1$, where KTF reduces to univariate trend filtering). As it turns out, the singular values can be handled with a relatively straightforward application of the Cauchy interlacing theorem. It is establishing the incoherence of the singular vectors that proves to be the real challenge. This is accomplished by leveraging specialized approximation bounds for the eigenvectors of Toeplitz matrices from [2].

**Theorem 3.** *Assume that $d = 2$. For $k = 0$, since KTF reduces to the GTF with $k = 0$ (and to TV denoising), it satisfies the result stated in the first part of Theorem 2.*

*For any integer $k \geq 1$, $C_n = \|\widetilde{\Delta}^{(k+1)}\theta_0\|_1$ and $\lambda \asymp n^{\frac{k}{k+2}}(\log n)^{\frac{1}{k+2}}C_n^{-\frac{k}{k+2}}$, the KTF estimator in (5), (9) satisfies*

$$\mathrm{MSE}(\hat{\theta}, \theta_0) = O_{\mathbb{P}}\left(\frac{1}{n} + n^{-\frac{2}{k+2}}(\log n)^{\frac{1}{k+2}}C_n^{\frac{2}{k+2}}\right).$$

The results in Theorems 2 and 3 match, in terms of their dependence on $n, k, d$ and the smoothness parameter $C_n$. As we will see in the next section, the smoothness classes defined by the GTF and KTF operators are similar, though not exactly the same, and each GTF and KTF is minimax rate optimal with respect to its own smoothness class, up to log factors.

**Beyond 2d?** To analyze GTF and KTF on grids of dimension $d \geq 3$, we would need to establish incoherence of the left singular vectors of the GTF and KTF operators. This should be possible by extending the arguments given in [31] (for GTF) and in the proof of Theorem 3 (for KTF), and is left to future work.

## 4 Lower bounds on estimation error

We present lower bounds on the minimax estimation error over smoothness classes defined by the operators from GTF (6) and KTF (9), denoted

$$\mathcal{T}_d^k(C_n) = \{\theta \in \mathbb{R}^n : \|\Delta^{(k+1)}\theta\|_1 \leq C_n\}, \tag{12}$$

$$\widetilde{\mathcal{T}}_d^k(C_n) = \{\theta \in \mathbb{R}^n : \|\widetilde{\Delta}^{(k+1)}\theta\|_1 \leq C_n\}, \tag{13}$$

respectively (where the subscripts mark the dependence on the dimension $d$ of the underlying grid graph). Before we derive such lower bounds, we examine embeddings of (the discretization of) the class of Holder smooth functions into the GTF and KTF classes, both to understand the nature of these new classes, and to define what we call a "canonical" scaling for the radius parameter $C_n$.

**Embedding of Holder spaces and canonical scaling.** Given an integer $k \geq 0$ and $L > 0$, recall that the *Holder class* $H(k + 1, L; [0, 1]^d)$ contains $k$ times differentiable functions $f : [0, 1]^d \to \mathbb{R}$, such that for all integers $\alpha_1, \ldots, \alpha_d \geq 0$ with $\alpha_1 + \cdots + \alpha_d = k$,

$$\left|\frac{\partial^k f(x)}{\partial x_1^{\alpha_1} \cdots \partial x_d^{\alpha_d}} - \frac{\partial^k f(z)}{\partial x_1^{\alpha_1} \cdots \partial x_d^{\alpha_d}}\right| \leq L\|x - z\|_2, \quad \text{for all } x, z \in [0, 1]^d.$$

To compare Holder smoothness with the GTF and KTF classes defined in (12), (13), we discretize the class $H(k + 1, L; [0, 1]^d)$ by considering function evaluations over the grid $Z_d$, defining

$$\mathcal{H}_d^{k+1}(L) = \left\{\theta \in \mathbb{R}^n : \theta(x) = f(x), \ x \in Z_d, \ \text{for some } f \in H(k + 1, L; [0, 1]^d)\right\}. \tag{14}$$

Now we ask: how does the (discretized) Holder class in (14) compare to the GTF and KTF classes in (12), (13)? Beginning with a comparison to KTF, fix $\theta \in \mathcal{H}_d^{k+1}(L)$, corresponding to evaluations of $f \in H(k + 1, L; [0, 1]^d)$, and consider a point $x \in Z_d$ that is away from the boundary. Then the KTF penalty at $x$ is

$$\begin{aligned}
\left|\left(D_{x_j^{k+1}}\theta\right)(x)\right| &= \left|\left(D_{x_j^k}\theta\right)(x + e_j/N) - \left(D_{x_j^k}\theta\right)(x)\right| \\
&\leq N^k\left|\frac{\partial^k}{\partial x_j^k}f(x + e_j/N) - \frac{\partial^k}{\partial x_j^k}f(x)\right| + N^k\delta(N) \\
&\leq LN^{k-1} + cLN^{k-1}. \tag{15}
\end{aligned}$$

In the second line above, we define $\delta(N)$ to be the sum of absolute errors in the discrete approximations to the partial derivatives (i.e., the error in approximating $\partial^k f(x)/\partial x_j^k$ by $(D_{x_j^k}\theta)(x)/N^k$, and similarly at $x + e_j/N$). In the third line, we use Holder smoothness to upper bound the first term, and we use standard numerical analysis (details in the supplement) for the second term to ensure that $\delta(N) \leq cL/N$ for a constant $c > 0$ depending only on $k$. Summing the bound in (15) over $x \in Z_d$ as appropriate gives a uniform bound on the KTF penalty at $\theta$, and leads to the next result.

**Lemma 3.** *For any integers $k \geq 0$, $d \geq 1$, the (discretized) Holder and KTF classes defined in* (14)*,* (13) *satisfy $\mathcal{H}_d^{k+1}(L) \subseteq \widetilde{\mathcal{T}}_d^k(cLn^{1-(k+1)/d})$, where $c > 0$ is a constant depending only on $k$.*

This lemma has three purposes. First, it provides some supporting evidence that the KTF class is an interesting smoothness class to study, as it shows the KTF class contains (discretizations of) Holder smooth functions, which are a cornerstone of classical nonparametric regression theory. In fact, this containment is strict and the KTF class contains more heterogeneous functions in it as well. Second, it leads us to define what we call the *canonical scaling $C_n \asymp n^{1-(k+1)/d}$* for the radius of the KTF class (13). This will be helpful for interpreting our minimax lower bounds in what follows; at this scaling, note that we have $\mathcal{H}_d^{k+1}(1) \subseteq \widetilde{\mathcal{T}}_d^k(C_n)$. Third and finally, it gives us an easy way to establish lower bounds on the minimax estimation error over KTF classes, by invoking well-known results on minimax rates for Holder classes. This will be described shortly.

As for GTF, calculations similar to (15) are possible, but complications ensue for $x$ on the boundary of the grid $Z_d$. Importantly, unlike the KTF penalty, the GTF penalty includes discrete derivatives at the boundary and so these complications have serious consequences, as stated next.

**Lemma 4.** *For any integers $k, d \geq 1$, there are elements in the (discretized) Holder class $\mathcal{H}_d^{k+1}(1)$ in* (14) *that do not lie in the GTF class $\mathcal{T}_d^k(C_n)$ in* (12) *for arbitrarily large $C_n$.*

This lemma reveals a very subtle drawback of GTF caused by the use of discrete derivatives at the boundary of the grid. The fact that GTF classes do not contain (discretized) Holder classes makes them seem less natural (and perhaps, in a sense, less interesting) than KTF classes. In addition, it means that we cannot use standard minimax theory for Holder classes to establish lower bounds for the estimation error over GTF classes. However, as we will see next, we can construct lower bounds for GTF classes via another (more purely geometric) embedding strategy; interestingly, the resulting rates match the Holder rates, suggesting that, while GTF classes do not contain all (discretized) Holder functions, they do contain "enough" of these functions to admit the same lower bound rates.

**Minimax rates for GTF and KTF classes.** Following from classical minimax theory for Holder classes [14, 29], and Lemma 3, we have the following result for the minimax rates over KTF classes.

**Theorem 4.** *For any integers $k \geq 0$, $d \geq 1$, the KTF class defined in* (13) *has minimax estimation error satisfying*

$$R\big(\widetilde{\mathcal{T}}_d^k(C_n)\big) = \Omega(n^{-\frac{2d}{2k+2+d}} C_n^{\frac{2d}{2k+2+d}}).$$

For GTF classes, we use a different strategy. We embed an ellipse, then rotate the parameter space and embed a hypercube, leading to the following result.

**Theorem 5.** *For any integers $k \geq 0$, $d \geq 1$, the GTF class defined in* (12) *has minimax estimation error satisfying*

$$R\big(\mathcal{T}_d^k(C_n)\big) = \Omega(n^{-\frac{2d}{2k+2+d}} C_n^{\frac{2d}{2k+2+d}}).$$

Several remarks are in order.

**Remark 2.** Plugging in the canonical scaling $C_n \asymp n^{1-(k+1)/d}$ in Theorems 4 and 5, we see that

$$R(\widetilde{\mathcal{T}}_d^k(C_n)) = \Omega(n^{-\frac{2k+2}{2k+2+d}}) \quad \text{and} \quad R(\mathcal{T}_d^k(C_n)) = \Omega(n^{-\frac{2k+2}{2k+2+d}}),$$

both matching the usual rate for the Holder class $\mathcal{H}_d^{k+1}(1)$. For KTF, this should be expected, as its lower bound is constructed via the Holder embedding given in Lemma 3. But for GTF, it may come as somewhat of a surprise—despite the fact it does not embed a Holder class as in Lemma 4, we see that the GTF class shares the same rate, suggesting it still contains something like "hardest" Holder smooth signals.

**Remark 3.** For $d = 2$ and all $k \geq 0$, we can certify that the lower bound rate in Theorem 4 is tight, modulo log factors, by comparing it to the upper bound in Theorem 3. Likewise, we can certify that the lower bound rate in Theorem 5 is tight, up to log factors, by comparing it to the upper bound in Theorem 2. For $d \geq 3$, the lower bound rates in Theorems 4 and 5 will not be tight for some values of $k$. For example, when $k = 0$, at the canonical scaling $C_n \asymp n^{1-1/d}$, the lower bound rate (given by either theorem) is $n^{-2/(2+d)}$, however, [22] prove that the minimax error of the TV class scales (up to log factors) as $n^{-1/d}$ for $d \geq 2$, so we see there is a departure in the rates for $d \geq 3$.

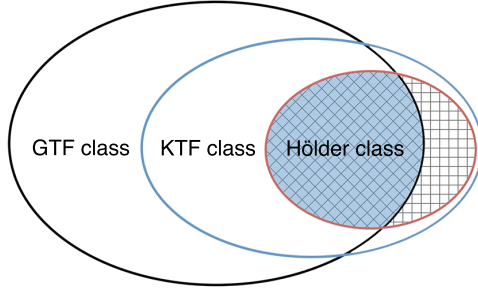

Figure 2: *Illustration of the two higher-order TV classes, namely the GTF and KTF classes, as they relate to the (discretized) Holder class. The horizontally/vertically checkered region denotes the part of Holder class not contained in the GTF class. As explained in Section 4, this is due to the fact that the GTF operator penalizes discrete derivatives on the boundary of the grid graph. The diagonally checkered region (also colored in blue) denotes the part of the Holder class contained in the GTF class. The minimax lower bound rates we derive for the GTF class in Theorem 5 match the well-known Holder rates, suggesting that this region is actually sizeable and contains the "hardest" Holder smooth signals.*

In general, we conjecture that the Holder embedding for the KTF class (and ellipse embedding for GTF) will deliver tight lower bound rates, up to log factors, when $k$ is large enough compared to $d$. This would have interesting implications for adaptivity to smoother signals (see the next remark); a precise study will be left to future work, along with tight minimax lower bounds for all $k, d$.

**Remark 4.** Again by comparing Theorems 3 and 4, as well as Theorems 2 and 5, we find that, for $d = 2$ and all $k \geq 0$, KTF is rate optimal for the KTF smoothness class and GTF is rate optimal for the GTF smoothness class, modulo log factors. We conjecture that this will continue to hold for all $d \geq 3$, which will be examined in future work. Moreover, an immediate consequence of Theorem 3 and the Holder embedding in Lemma 3 is that KTF adapts automatically to Holder smooth signals, i.e., it achieves a rate (up to log factors) of $n^{-(k+1)/(k+2)}$ over $\mathcal{H}_2^{k+1}(1)$, matching the well-known minimax rate for the more homogeneous Holder class. It is not clear that GTF shares this property.

## 5 Discussion

In this paper, we studied two natural higher-order extensions of the TV estimator on $d$-dimensional grid graphs. The first was graph trend filtering (GTF) as defined in [31], applied to grids; the second was a new Kronecker trend filtering (KTF) method, which was built with the special (Euclidean-like) structure of grids in mind. GTF and KTF exhibit some similarities, but are different in important ways. Notably, the notion of smoothness defined using the KTF operator is somewhat more natural, and is a strict generalization of the standard notion of Holder smoothness (in the sense that the KTF smoothness class strictly contains a Holder class of an appropriate order). This is not true for the notion of smoothness defined using the GTF operator. Figure 2 gives an illustration.

When $d = 2$, we derived tight upper bounds for the estimation error achieved by the GTF and KTF estimators—tight in the sense that these upper bound match in rate (modulo log factors) the lower bounds on the minimax estimation errors for the GTF and KTF classes. We constructed the lower bound for the KTF class by leveraging the fact that it embeds a Holder class; for the GTF class, we used a different (more geometric) embedding. While these constructions proved to be tight for $d = 2$ and all $k \geq 0$, we suspect this will no longer be the case in general, when $d$ is large enough relative to $k$. We will examine this in future work, along with upper bounds for GTF and KTF when $d \geq 3$.

Another important consideration for future work are the minimax linear rates over GTF and KTF classes, i.e., minimax rates when we restrict our attention to linear estimators. We anticipate that a gap will exist between minimax linear and nonlinear rates for all $k, d$ (as it does for $k = 0$, as shown in [22]). This would, e.g., provide some rigorous backing to the claim that the KTF class is larger than its embedded Holder class (the latter having matching minimax linear and nonlinear rates).

**Acknowledgements.** We thank Sivaraman Balakrishnan for helpful discussions regarding minimax rates for Holder classes on grids. JS was supported by NSF Grant DMS-1712996. VS, YW, and RT were supported by NSF Grants DMS-1309174 and DMS-1554123.

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
