[Supplementary Material · supp_nips.pdf]

# Supplement to "Higher-Order Total Variation Classes on Grids: Minimax Theory and Trend Filtering Methods"

**Veeranjaneyulu Sadhanala**
Carnegie Mellon University
Pittsburgh, PA 15213
vsadhana@cs.cmu.edu

**Yu-Xiang Wang**
Carnegie Mellon University/Amazon AI
Pittsburgh, PA 15213/Palo Alto, CA 94303
yuxiangw@amazon.com

**James Sharpnack**
University of California, Davis
Davis, CA 95616
jsharpna@ucdavis.edu

**Ryan J. Tibshirani**
Carnegie Mellon University
Pittsburgh, PA 15213
ryantibs@stat.cmu.edu

We provide proofs and additional details for the results in "Higher-Order Total Variation Classes on Grids: Minimax Theory and Trend Filtering Methods".

## A.1 Proof of Lemma 1

The nullity of $\widetilde{\Delta}^{(k+1)}$ is the number of nonzero singular values of $\widetilde{\Delta}^{(k+1)}$, or equivalently, the number of nonzero eigenvalues of $(\widetilde{\Delta}^{(k+1)})^T \widetilde{\Delta}^{(k+1)}$. Following from (9), and abbreviating $D = D_{1d}^{(k+1)}$,

$$(\widetilde{\Delta}^{(k+1)})^T \widetilde{\Delta}^{(k+1)} = D^T D \otimes I \otimes \cdots \otimes I \; + \; I \otimes D^T D \otimes \cdots \otimes I \; + \; \ldots$$
$$+ \; I \otimes I \otimes \cdots \otimes D^T D,$$

the Kronecker sum of $D^T D$ with itself, a total of $d$ times. Using a standard fact about Kronecker sums, if $\rho_i, i = 1, \ldots, N$ denote the eigenvalues of $D^T D$ then

$$\rho_{i_1} + \rho_{i_2} + \cdots + \rho_{i_d}, \quad i_1, \ldots, i_d \in \{1, \ldots, N\}$$

are the eigenvalues of $(\widetilde{\Delta}^{(k+1)})^T \widetilde{\Delta}^{(k+1)}$. By counting the multiplicity of the zero eigenvalue, we arrive at a nullity for $\widetilde{\Delta}^{(k+1)}$ of $(k+1)^d$. It is straightforward to check that the vectors specified in the lemma, given by evaluations of polynomials, are in the null space, and that these are linearly independent, which completes the proof. □

## A.2 Proof of Lemma 2

Let us define

$$\widetilde{D} = \left[ \begin{array}{c} C_{1d}^{(k+1)} \\ D_{1d}^{(k+1)} \end{array} \right] \in \mathbb{R}^{N \times N},$$

where the first $k+1$ rows are given by a matrix $C^{(k+1)} \in \mathbb{R}^{(k+1) \times N}$ that completes the row space, as in Lemma 2 of Wang et al. [9]. And now, again by Lemma 2 of Wang et al. [9],

$$\left( H_{1d}^{(k)} \right)^{-1} = \frac{1}{k!} \widetilde{D}, \tag{A.1}$$

where $H_{1d}^{(k)} \in \mathbb{R}^{N \times N}$ is the falling factorial basis matrix of order $k$, which has elements

$$\left( H_{1d}^{(k)} \right)_{ij} = h_j(i/N), \quad i, j = 1, \ldots, N,$$

with $h_i$, $i = 1, \ldots, N$ denoting the falling factorial basis functions in (10).

Let us write the KTF problem in (5), (9) explicitly as

$$\min_{\theta \in \mathbb{R}^n} \frac{1}{2}\|y - \theta\|_2^2 + \lambda \left\| \begin{bmatrix} D_{1\mathrm{d}}^{(k+1)} \otimes I \otimes \cdots \otimes I \\ I \otimes D_{1\mathrm{d}}^{(k+1)} \otimes \cdots \otimes I \\ \vdots \\ I \otimes I \otimes \cdots \otimes D_{1\mathrm{d}}^{(k+1)} \end{bmatrix} \theta \right\|_1. \tag{A.2}$$

We now transform variables in this problem by defining $\theta = (H_{1\mathrm{d}}^{(k)} \otimes \cdots \otimes H_{1\mathrm{d}}^{(k)})\alpha$ and using (A.1), which turns (A.2) into an equivalent basis form,

$$\min_{\alpha \in \mathbb{R}^n} \frac{1}{2}\left\|y - \left(H_{1\mathrm{d}}^{(k)} \otimes \cdots \otimes H_{1\mathrm{d}}^{(k)}\right)\alpha\right\|_2^2 + \lambda k! \left\| \begin{bmatrix} I^0 \otimes H_{1\mathrm{d}}^{(k)} \otimes \cdots \otimes H_{1\mathrm{d}}^{(k)} \\ H_{1\mathrm{d}}^{(k)} \otimes I^0 \otimes \cdots \otimes H_{1\mathrm{d}}^{(k)} \\ \vdots \\ H_{1\mathrm{d}}^{(k)} \otimes H_{1\mathrm{d}}^{(k)} \otimes \cdots \otimes I^0 \end{bmatrix} \alpha \right\|_1, \tag{A.3}$$

where $I^0 = \begin{bmatrix} 0_{(N-k-1)\times(k+1)} & I_{(N-k-1)} \end{bmatrix}$.

Interestingly, the penalty in (A.3) is not a pure sparsity penalty on the coefficients $\alpha$ (as it is in basis form in 1d) but a sparsity penalty on aggregated (sums of) coefficients. This makes the penalty a little hard to interpret, but to glean intuition, we can rewrite the problem once more via the transformation

$$f = \sum_{i_1,\ldots,i_d=1}^{N} \alpha_{i_1,\ldots,i_d}(h_{i_1} \otimes h_{i_2} \otimes \cdots \otimes h_{i_d}), \tag{A.4}$$

where recall we are indexing the components of $\alpha$ by $\alpha_{i_1,\ldots,i_d}$, for $i_1, \ldots, i_d = 1, \ldots, N$ (and the summands above use tensor products of univariate functions). To be concrete, note that the function $f$ defined in (A.4) evaluates to

$$f(x) = \sum_{i_1,\ldots,i_d=1}^{N} \alpha_{i_1,\ldots,i_d} h_{i_1}(x) h_{i_2}(x_2) \cdots h_{i_d}(x_d), \quad x \in [0,1]^d.$$

Thus we can equivalently write the basis form in (A.3) in functional form

$$\min_{f \in H_d} \frac{1}{2} \sum_{x \in Z_d} \left(y(x) - f(x)\right)^2 + \lambda \sum_{j=1}^{d} \sum_{x_{-j} \in Z_{d-1}} \mathrm{TV}\left(\frac{\partial^k f(\cdot, x_{-j})}{\partial x_j^k}\right), \tag{A.5}$$

where recall $f(\cdot, x_{-j})$ denotes $f$ as function of the $j$th dimension with all other dimensions fixed at $x_{-j}$, $\partial^k/\partial x_j^k(\cdot)$ denotes the $k$th partial weak derivative operator with respect to $x_j$, for $j = 1, \ldots, d$, and $\mathrm{TV}(\cdot)$ denotes the total variation operator. To see the equivalence between the penalty terms in (A.3) and (A.5), it can be directly checked that

$$k!\left(I^0 \otimes H_{1\mathrm{d}}^{(k)} \otimes \cdots \otimes H_{1\mathrm{d}}^{(k)}\right)\alpha$$

contains the differences of the function $\partial^k f/\partial x_1^k$ over all pairs of grid positions that are adjacent in the $x_1$ direction, where $f$ is as in (A.4). This, combined with the fact that $\partial^k f/\partial x_1^k$ is constant in between lattice positions, means that

$$k!\left\|\left(I^0 \otimes H_{1\mathrm{d}}^{(k)} \otimes \cdots \otimes H_{1\mathrm{d}}^{(k)}\right)\alpha\right\|_1 = \sum_{x_{-1} \in Z_{d-1}} \mathrm{TV}\left(\frac{\partial^k f(\cdot, x_{-1})}{\partial x_1^k}\right),$$

the total variation of $\partial^k f/\partial x_1^k$ added up over all slices of the lattice $Z_d$ in the $x_1$ direction. Similar arguments apply to the penalty terms corresponding to dimensions $j = 2, \ldots, d$, and this completes the proof. $\square$

## A.3 Proof of Theorem 2

For $d = 2$, it is shown in the proof of Corollary 8 in Wang et al. [10] that the GTF operator $\Delta^{(k+1)}$ satisfies the incoherence property, as defined in Theorem 1, for any choice of cutoff $i_0 \geq 1$, and with a constant $\mu = 4$ when $k$ is even and $\mu = 2$ when $k$ is odd. It suffices to upper bound the partial sum term $\sum_{i=i_0+1}^{n-1} \xi_i^{-2}$. Lemma A.1 gives the key calculation, where it is shown that for large enough $n$ and each $i_0 \geq 1$,

$$\sum_{i=i_0+1}^{n-1} \frac{1}{\xi_i^2} \leq c \cdot \begin{cases} n\log(n/i_0) & \text{for } k = 0 \\ n^{k+1} i_0^{-k} & \text{for } k \geq 1, \end{cases}$$

where $c > 0$ is a constant that depends only on $k$. For $k = 0$, to minimize to the upper bound given in Theorem 1, we want to balance

$$\frac{i_0}{n} \quad \text{with} \quad \frac{\mu}{n}\sqrt{\log n \log(n/i_0)}C_n.$$

This leads us to choose $i_0 \asymp C_n \log n$, and plugging this in gives the result for $k = 0$.

For $k \geq 1$, we want to balance

$$\frac{i_0}{n} \quad \text{with} \quad \frac{\mu}{n}\sqrt{\log n (n/i_0)^k}C_n.$$

This leads us to take $i_0 \asymp n^{k/(k+2)}(\log n)^{1/(k+2)}C_n^{2/(k+2)}$, and plugging this in completes the proof for $k \geq 1$. □

## A.4 Lemma A.1

The next lemma is the key driver for the sharp rate established in Theorem 2. Here and henceforth, denote $[i] = \{1, \ldots, i\}$ for an integer $i \geq 1$.

**Lemma A.1.** *Let $\xi_1 \leq \ldots \leq \xi_{n-1}$ be the nonzero singular values of the GTF operator $\Delta^{(k+1)}$ of order $k + 1$. If $k = 0$, then for any $i_0 \in [n-1]$,*

$$\sum_{i=i_0+1}^{n-1} \frac{1}{\xi_i^2} \leq cn \log(n/i_0).$$

*for large enough $n$, where $c > 0$ is an absolute constant. If $k > 1$, then for any $i_0 \in [n-1]$,*

$$\sum_{i=i_0+1}^{n-1} \frac{1}{\xi_i^2} \leq cn^{k+1}/i_0^k,$$

*for large enough $n$, where now $c > 0$ is a constant depending only on $k$.*

*Proof.* In the following, we denote by $c > 0$ a constant whose value may change from line to line, as needed.

Let us denote by $\lambda_1 \leq \ldots \leq \lambda_{n-1}$ the nonzero eigenvalues of the Laplacian of the 2d grid graph of size $N \times N$. As shown in Wang et al. [10], the GTF operator $\Delta^{(k+1)}$ has squared singular values $\xi_i^2 = \lambda_i^{k+1}$, $i \in [n-1]$. We can index the eigenvalues of the Laplacian by 2d grid positions, and we note (as, e.g., in the proof of Corollary 8 in Wang et al. [10]) that they may be written as

$$\lambda_{i_1,i_2} = 4\sin^2\left(\frac{\pi(i_1-1)}{2N}\right) + 4\sin^2\left(\frac{\pi(i_2-1)}{2N}\right), \quad i_1, i_2 \in [N].$$

For the first claim in the lemma, take $j_0 = \lfloor\sqrt{i_0}\rfloor$. Observe, using $\sin(x) \geq x/2$ for $x \in [0, \pi/2]$,

$$\sum_{i=i_0+1}^{n-1} \frac{1}{\lambda_i} \leq \sum_{\min\{i_1,i_2\}\geq j_0+1} \frac{1}{\lambda_{i_1,i_2}}$$

$$\leq cn \sum_{\min\{i_1,i_2\}\geq j_0+1} \frac{1}{(i_1-1)^2 + (i_2-1)^2}$$

$$\leq cn \sum_{i_1=j_0}^{N-1} \sum_{i_2=1}^{N-1} \frac{1}{i_1^2 + i_2^2}$$

$$\leq cn \sum_{i_1=j_0}^{N-1} \int_0^{N-1} \frac{1}{i_1^2 + x^2} \, dx$$

$$= cn \sum_{i_1=j_0}^{N-1} \frac{1}{i_1} \tan^{-1}\left(\frac{N-1}{i_1}\right)$$

$$\leq cn \sum_{i_1=j_0}^{N-1} \frac{1}{i_1} \frac{\pi}{2}$$

$$\leq cn \log(N/j_0),$$

for sufficiently large $n$.

As for the second claim in the lemma, observe, again using $\sin(x) \geq x/2$ for $x \in [0, \pi/2]$,

$$\sum_{i=i_0+1}^{n-1} \frac{1}{\lambda_i^{k+1}} \leq \sum_{(i_1-1)^2+(i_2-1)^2 \geq i_0}^{n} \frac{1}{\lambda_{i_1,i_2}^{k+1}}$$

$$\leq cn^{k+1} \sum_{(i_1-1)^2+(i_2-1)^2 \geq i_0} \frac{1}{((i_1-1)^2 + (i_2-1)^2)^{k+1}}$$

$$\leq cn^{k+1} \left( \int_{i_0 \leq x^2+y^2 \leq 2(n-1),\, x,y \geq 0} \frac{1}{(x^2+y^2)^{k+1}} \, dx \, dy + \sum_{(i_1-1)^2+(i_2-1)^2=i_0} \frac{1}{i_0^{k+1}} \right)$$

$$\leq cn^{k+1} \left( \int_0^{\pi/2} \int_{\sqrt{i_0}}^{\sqrt{2(n-1)}} \frac{1}{r^{2(k+1)}} r \, dr \, d\theta + \frac{1}{i_0^{k+1/2}} \right)$$

$$\leq cn^{k+1} \left( \frac{\pi}{2} \int_{i_0}^{2(n-1)} \frac{1}{u^{k+1}} \, du + \frac{1}{i_0^{k+1/2}} \right)$$

$$= cn^{k+1} \left( \frac{\pi}{2} \left( \frac{1}{i_0^k} - \frac{1}{(2(n-1))^k} \right) + \frac{1}{i_0^{k+1/2}} \right)$$

$$\leq cn^{k+1}/i_0^k.$$

$\square$

## A.5   Proof of Theorem 3

The KTF operator (9), in the current case that $d = 2$, is simply

$$\widetilde{\Delta}^{(k+1)} = \left[ \begin{array}{c} D_{\text{1d}}^{(k+1)} \otimes I \\ I \otimes D_{\text{1d}}^{(k+1)} \end{array} \right].$$

Abbreviate $N' = N - k - 1$. Let $\beta_i, u_i, v_i$ be a triplet of nonzero singular value, left singular vector, and right singular vector of $D_{\text{1d}}^{(k+1)}$, for $i \in [N']$. We seek the left singular values of $\widetilde{\Delta}^{(k+1)}$, i.e, the eigenvectors (corresponding to nonzero eigenvalues) of

$$\widetilde{\Delta}^{(k+1)} (\widetilde{\Delta}^{(k+1)})^T = \left[ \begin{array}{cc} DD^T \otimes I & D \otimes D^T \\ D^T \otimes D & I \otimes DD^T \end{array} \right],$$

where we abbreviate $D = D_{\text{1d}}^{(k+1)}$. The vectors

$$\left[ \begin{array}{c} \beta_i \cdot u_i \otimes v_j \\ \beta_j \cdot v_i \otimes u_j \end{array} \right], \quad i, j \in [N'], \tag{A.6}$$

for $i, j \in [N']$ are $(N')^2$ such eigenvectors. The other eigenvectors are given by

$$
\begin{bmatrix} u_i \otimes p_j \\ 0 \end{bmatrix}, \quad \begin{bmatrix} 0 \\ p_j \otimes u_i \end{bmatrix}, \quad i \in [N'], j \in [k+1], \tag{A.7}
$$

where $p_j$, $j \in [k+1]$ form an orthogonal basis for the null space of $D_{1d}^{(k+1)}$.

The main technical challenge is in establishing incoherence of the vectors in (A.6), (A.7). We note that, based on the Kronecker product form of these vectors, it suffices to show incoherence of $u_i, v_i$, $i \in [N']$, and $p_i$, $i \in [k+1]$. Incoherence of $u_i$, $i \in [N']$ is established in Lemma A.3 and of $v_i$, $i \in [N']$ in Lemma A.4, using specialized approximations for eigenvectors of Toeplitz matrices from Bogoya et al. [1]. Incoherence of $p_i$, $i \in [k+1]$ may be seen by choosing, e.g., these vectors to be the discrete Legendre orthogonal polynomials as in Neuman and Schonbach [5]. We have thus shown that $\widetilde{\Delta}^{(k+1)}$ satisfies the incoherence property, as defined in Theorem 1, for any choice of $i_0 \geq 1$.

Now we address the partial sum term $\sum_{i=i_0+1}^{n-1} \xi_i^{-2}$. Lemma A.2 shows that for large enough $n$ and a constant $c > 0$ depending only on $k$,

$$
\sum_{i=i_0+1}^{n-1} \frac{1}{\xi_i^2} \leq c \cdot \begin{cases} n \log(n/i_0) & \text{for } k = 0 \\ n^{k+1} i_0^{-k} & \text{for } k \geq 1, \end{cases}
$$

just as was the case for GTF. (In fact, this result is proved by tying the singular values of the KTF operator to those of the GTF operator.) Repeating the same arguments as in the proof of Theorem 2 gives the desired result. □

## A.6 Lemma A.2

This lemma provides a result analogous to Lemma A.1, by tying together the singular values of the KTF and GTF operators.

**Lemma A.2.** *Let $\xi_1 \leq \ldots \leq \xi_{n-(k+1)^2}$ be the nonzero singular values of the KTF operator $\widetilde{\Delta}^{(k+1)}$ of order $k + 1$. If $k = 0$, then for any $i_0 \in [n - (k+1)^2 - 1]$,*

$$
\sum_{i=i_0+1}^{n-(k+1)^2} \frac{1}{\xi_i^2} \leq cn \log(n/i_0).
$$

*for large enough $n$, where $c > 0$ is an absolute constant. If $k > 1$, then for $i_0 \in [n - (k+1)^2 - 1]$,*

$$
\sum_{i=i_0+1}^{n-(k+1)^2} \frac{1}{\xi_i^2} \leq cn^{k+1}/i_0^k,
$$

*for large enough $n$, where now $c > 0$ is a constant depending only on $k$.*

*Proof.* Abbreviate $D = D_{1d}^{(k+1)}$, and write $G$ for the GTF operator of order $k + 1$ defined over a 1d chain of length $N$. Also let $N' = N - k - 1$, and $k' = \lfloor (k+1)/2 \rfloor$. Then $D$ is given by removing the first $k_1$ rows and last $k_2$ rows of $G$, i.e.,

$$
D = PG, \quad \text{where } P = \begin{bmatrix} 0_{N' \times k'} & I_{N'} & 0_{N' \times k'} \end{bmatrix}.
$$

This means

$$
DD^T = PGG^T P^T.
$$

Let $\beta_i$, $i \in [N']$ be the eigenvalues of $DD^T$, and let $\alpha_i$, $i \in [N]$ be the eigenvalues of $GG^T$. The Cauchy interlacing theorem now tells us that

$$
\beta_i \geq \alpha_i^{k+1}, \quad i \in [N']. \tag{A.8}
$$

This key property will allow us to relate the nonzero singular values of the KTF operator to those of the GTF operator, more specifically, to the eigenvalues of the Laplacian of the 2d grid graph.

The squared nonzero singular values of $\widetilde{\Delta}^{(k+1)}$ are the nonzero eigenvalues of $(\widetilde{\Delta}^{(k+1)})^T \widetilde{\Delta}^{(k+1)}$. We can index the eigenvalues of $(\widetilde{\Delta}^{(k+1)})^T \widetilde{\Delta}^{(k+1)}$ by 2d grid positions, as in

$$
\psi_{i_1,i_2} = \rho_{i_1} + \rho_{i_2}, \quad i_1, i_2 \in [N],
$$

where $\rho_i$, $i \in [N]$ denote the eigenvalues of $D^T D$, i.e., $\rho_1 = \cdots = \rho_{k+1} = 0$ and $\rho_{i+k+1} = \beta_i$, $i \in [N']$, where $D$, $\beta_i$, $i \in [N']$ are as above. Also, as in the proof of Lemma A.1, we can write the eigenvalues of the Laplacian matrix of the 2d grid graph as

$$\lambda_{i_1, i_2} = \alpha_{i_1} + \alpha_{i_2}, \quad i_1, i_2 \in [N]$$

where $\alpha_i$, $i \in [N]$ is as above. For arbitrary $i_1, i_2 \in [N]$, with at least one $i_1 > k+2$ or $i_2 > k+2$, note that

$$\frac{1}{\psi_{i_1, i_2}} = \frac{1}{\beta_{i_1-k-1} + \beta_{i_2-k-1}} \leq \frac{1}{\alpha_{i_1-k-1}^{k+1} + \alpha_{i_2-k-1}^{k+1}} \leq \frac{2^{k+1}}{\lambda_{i_1-k-1, i_2-k-1}^{k+1}},$$

where we use the convention $\beta_{-i} = 0$ and $\alpha_{-i} = 0$ for $i \leq 0$, the first inequality was due to the key property (A.8), and the second was due to the simple fact $(a+b)^k \leq 2^k a^k + 2^k b^k$. The last display shows that to bound the sum of squared reciprocal nonzero singular values of the KTF operator, it suffices to bound the reciprocal $k$th power of Laplacian eigenvalues, as was the case for the GTF operator. Proceeding as in the proof of Lemma A.1 gives the result. $\qquad\square$

## A.7 Lemmas A.3, A.4, and A.5

In this section, the first two lemmas establish incoherence of the left and right singular vectors of $D_{1d}^{(k+1)}$. They rely heavily on approximation results for eigenvectors of symmetric banded Toeplitz matrices in Bogoya et al. [1]. The third lemma relates the eigenvectors of $(D_{1d}^{(k+1)})(D_{1d}^{(k+1)})^T$ to those of its elementwise absolute value matrix. This is critical for the proof of the first lemma, since, curiously, $(D_{1d}^{(k+1)})(D_{1d}^{(k+1)})^T$ falls outside of the scope of matrices considered in Bogoya et al. [1] (as well as related papers on eigenvector approximations for Toeplitz matrices), but the elementwise absolute value matrix does not.

**Lemma A.3.** *The left singular vectors $u_i$, $i \in [N-k-1]$ of $D_{1d}^{(k+1)} \in \mathbb{R}^{N \times (N-k-1)}$ are incoherent, i.e., there exists a constant $\mu > 0$ depending only on $k$ such that*

$$\|u_i\|_\infty \leq \frac{\mu}{\sqrt{N}}, \quad i \in [N-k-1],$$

*for a constant $\mu > 0$ depending only on $k$.*

*Proof.* For $k = 0$, the result has already been proved in Wang et al. [10]. Assume $k \geq 1$ henceforth. As in Lemma A.5 and its proof, abbreviate $D = D_{1d}^{(k+1)}$, and $N' = N - k - 1$. The left singular vectors of $D$ are the eigenvectors of $DD^T$, which is a symmetric banded Toeplitz matrix with entries

$$(DD^T)_{ij} = c_{|i-j|}, \quad i, j \in [N'],$$
$$\text{where} \quad c_\ell = (-1)^\ell \binom{2k+2}{k+1+\ell}, \quad \ell = 0, \ldots, k+1.$$

Let $\beta_1 \leq \ldots \leq \beta_{N'}$ be the eigenvalues of $DD^T$. Observe that $\beta_{N'} \leq 4^{k+1}$ by the Gershgorin circle theorem.

Unfortunately, the approximation results on eigenvectors of Toeplitz matrices from Bogoya et al. [1] are not applicable to $DD^T$, because $DD^T$ does not satisfy their simple-loop assumption. However, the Toeplitz matrix

$$T = 4^{k+1} I - \text{abs}(DD^T),$$

where $\text{abs}(A)$ denotes the elementwise absolute value of a matrix $A$, does satisfy the simple-loop assumption, and its eigenvectors are the same as those of $DD^T$ up to elementwise sign flips, as we show in Lemma A.5. Thus, it suffices to verify the incoherence property for $T$, which we pursue in the following.

To be concrete, $T$ is a symmetric banded Toeplitz matrix with entries

$$T_{ij} = a_{|i-j|}, \quad i, j \in [N'],$$
$$\text{where} \quad a_\ell = 4^{k+1} \cdot 1\{\ell = 0\} - \binom{2k+2}{k+1+\ell}, \quad \ell = 0, \ldots, k+1.$$

We introduce some notation. Let $\mathbb{C}$ denote the complex plane and $\mathbb{T}$ the unit circle in $\mathbb{C}$. The symbol of $T$ is the function $a : \mathbb{T} \to \mathbb{C}$ is defined by

$$a(t) = \sum_{\ell=-(k+1)}^{k+1} a_\ell t^\ell = 4^{k+1} - \left(2 + t + \frac{1}{t}\right)^{k+1}.$$

We define the function $g : [0, 2\pi) \to \mathbb{R}$ by $g(\sigma) = a(e^{\iota\sigma}) = 4^{k+1} - (2 + 2\cos\sigma)^{k+1}$. (Here we use $\iota = \sqrt{-1}$ for the imaginary unit, to differentiate it from the the index variable $i$.)

It is straightforward to check that $a, g$ as defined above satisfy what Bogoya et al. [1] refer to as the "simple-loop" conditions: $a$ is real-valued, the range of $g$ is contained in the bounded set $[0, 4^{k+1}]$, $g$ satisfies $g(0) = g(2\pi) = 0$, $g''(0) = g''(2\pi) > 0$, and $g$ reaches its maximum of $4^{k+1}$ at $\pi \in [0, 2\pi)$. Hence, in the notation of Bogoya et al. [1], we have $a \in SL^\alpha$ for any $\alpha \geq 4$.

For an eigenvalue $\tau$, the characteristic polynomial of $T$ is given by

$$p_\tau(t) = a(t) - \tau,$$

whose $2k+2$ are denoted by $z_0(\tau), z_1(\tau), z_2(\tau), \ldots, z_k(\tau)$ and their inverses. Following Bogoya et al. [1], we use a labeling convention such that $|z_0(\tau)| = 1$, and $|z_\kappa(\tau)| > 1$ for $\kappa \in [k]$. We also define the function $b : \mathbb{T} \times [0, \pi] \to (0, \infty)$ by

$$b(t, s) = \frac{a(t) - g(s)}{2\cos s - (1 + 1/t)} = \frac{(2 + t + 1/t)^{k+1} - (2 + 2\cos\sigma)^{k+1}}{(2 + 2\cos s) - (2 + t + 1/t)}.$$

(Here we are using the simplified form of $b$ in Corollary 2.2 of Bogoya et al. [1], due to symmetry of $g$.) As $b$ is a rational function (ratio of two polynomials) in $(t, s)$, denoted

$$b(t, s) = \frac{P(t, s)}{Q(t, s)}$$

it has a Wiener-Hopf factorization $b(t, s) = b_-(t, s)b_+(t, s)$, where

$$b_+(t, s) = b_0(s)\frac{\prod_{i=1}^{p}(1 - t/\nu_i(s))}{\prod_{i=1}^{q}(1 - t/\zeta_i(s))}$$

for a constant $b_0(s)$, where $\nu_i(s)$, $i \in [p]$ and $\zeta_i(s)$, $i \in [q]$ denote the roots of $P(\cdot, s)$ and $Q(\cdot, s)$, respectively, with complex moduli larger at least 1. (The term $b_-(t, s)$ has a similar representation, but the specific details are unimportant for our purposes.)

Because $a(t) - g(s)$ is the characteristic polynomial $p_\tau(t)$ with $\tau = g(s)$, the roots $\nu_i(s)$, $i \in [p]$ of $P(\cdot, s)$ are simply $z_0(g(s))$, $z_\kappa(g(s))$, $\kappa \in [k]$; moreover, according to Chapter 1 in Bottcher and Grudsky [2], the positive Wiener-Hopf factor $b_+(t, s)$ in the last display can be simplified to

$$b_+(t, s) = \prod_{\kappa=1}^{k} \left(t - z_\kappa(g(s))\right). \tag{A.9}$$

We are now ready to state the eigenvector approximation result. Write $\tau_i, \tilde{u}_i$ for a pair of eigenvalue and (unit norm) eigenvector of $T$, for $i \in [N']$. Combining Theorem 2.5, Theorem 4.1, and Lemma 4.2 in Bogoya et al. [1], for each $i \in [N']$, we can represent $\tilde{u}_i = e_i/\|e_i\|_2$, where

$$e_i = M_i + L_i + R_i + \delta_i, \tag{A.10}$$

and for each $j \in [N']$,

$$M_{ij} = \frac{z_{0i}^{\frac{N'-1}{2}-j+1}}{|b_{+i}(z_{0i})|} + (-1)^{N'-i}\frac{\bar{z}_{0i}^{\frac{N'-1}{2}-j+1}}{|b_{+i}(\bar{z}_{0i})|}, \tag{A.11}$$

$$L_{ij} = \frac{z_{0i}^{\frac{N'+1}{2}}(z_{0i} - \bar{z}_{0i})b_{+i}(z_{0i})}{|b_{+i}(z_{0i})|} \sum_{\kappa=1}^{k} \frac{z_\kappa(\tau_i)^{-j}}{\frac{\partial b_{+i}}{\partial t}(z_\kappa(\tau_i))(z_\kappa(\tau_i) - z_{0i})(z_\kappa(\tau_i) - \bar{z}_{0i})} \tag{A.12}$$

$$R_{ij} = \bar{L}_{i,N'+1-j}. \tag{A.13}$$

Here, $\delta_{ij} = o(1/N')$, uniformly over $i, j$, and we use the abbreviations $b_{+i}(t) = b_+(t, s_i)$, where $s_i$ is such that $g(s_i) = \tau_i$, and $z_{0i} = z_0(\tau_i)$, $i \in [N']$.

The details of the approximation in (A.10)–(A.13) are important for the next lemma, Lemma A.4, but are not needed presently. By the triangle inequality, for each $i, j \in [N']$,

$$\frac{|e_{ij}|}{\|e_i\|_2} \leq \frac{|M_{ij}|}{\|e_i\|_2} + \frac{|e_{ij} - M_{ij}|}{\|e_i\|_2} \leq \frac{1/|b_{+i}(z_{0i})| + 1/|b_{+i}(\bar{z}_{0i})|}{\|e_i\|_2} + \frac{|e_{ij} - M_{ij}|}{\|e_i\|_2}, \tag{A.14}$$

the second inequality following as $|z_{0i}| = 1$. Furthermore, by Theorem 2.6 and Lemma 4.2 in Bogoya et al. [1], we know that for each $i \in [N']$,

$$\|e_i\|_2 = \sqrt{N'}\Big(b_{+i}(z_{0i})^{-2} + b_{+i}(\bar{z}_{0i})^{-2}\Big)^{1/2} + O(1), \quad \text{and} \tag{A.15}$$

$$\frac{\|e_i - M_i\|_2}{\|e_i\|_2} = O\Big(\frac{1}{\sqrt{N'}}\Big), \tag{A.16}$$

where the $O(1), O(1/\sqrt{N'})$ terms in the above are uniform over $i$. Their Theorem 2.6 also shows that $(b_{+i}(z_{0i})^{-2} + b_{+i}(\bar{z}_{0i})^{-2})^{1/2} \asymp 1$, uniformly over $i$. Noting the equivalence of $\ell_1$ and $\ell_2$ norms in $\mathbb{R}^2$, we also have that $|b_{+i}(z_{0i})| + |b_{+i}(\bar{z}_{0i})| \asymp 1$, uniformly over $i$, and therefore, combining this with (A.14)–(A.16), we conclude

$$|\tilde{u}_{ij}| = \frac{|e_{ij}|}{\|e_i\|_2} \leq O\Big(\frac{1}{\sqrt{N'}}\Big),$$

uniformly over $i, j \in [N']$, which completes the proof. $\qquad\square$

**Lemma A.4.** *The right singular vectors $v_i$, $i \in [N-k-1]$ of $D_{1d}^{(k+1)} \in \mathbb{R}^{N \times (N-k-1)}$ are incoherent, i.e., there exists a constant $\mu > 0$ depending only on $k$ such that*

$$\|v_i\|_\infty \leq \frac{\mu}{\sqrt{N}}, \quad i \in [N - k - 1],$$

*for a constant $\mu > 0$ depending only on $k$.*

*Proof.* As before, abbreviate $D = D_{1d}^{(k+1)}$, and $N' = N - k - 1$. Denote by $\beta_i, u_i, v_i$ a triplet of nonzero singular value, left singular vector, and right singular vector of $D$, for $i \in [N']$. Also denote by $\tilde{u}_i$, $i \in [N']$ the eigenvectors of $T = 4^{k+1}I - \mathrm{abs}(DD^T)$.

Note that by Lemma A.5 we have the relationship

$$u_i = S\tilde{u}_i, \quad i \in [N'], \tag{A.17}$$

between the left singular vectors of $D$ and eigenvectors of $T$, where $S$ is the alternating sign diagonal matrix (as defined in the proof of the lemma). Note also the relationship

$$\sqrt{\beta_i}v_i = D^T u_i, \quad i \in [N'], \tag{A.18}$$

between the right and left singular vectors of $D$. We will bound the absolute entries of $v_i$, $i \in [N']$ over the interior and boundary coordinates separately.

**Bounding the interior elements.** Using (A.17), (A.18), we can translate the expansion in (A.10) for $\tilde{u}_i = e_i/\|e_i\|_2, i \in [N']$ into one for $v_i, i \in [N']$. Write $w_i = D_{1i}, i \in [k+2]$ for the $(k+1)$st order forward difference coefficients. Fix an arbitrary $i \in [N']$ and interior coordinate $j \in \{k+2, \ldots, N'\}$. We have, abbreviating $j' = j - k - 2$,

$$\sqrt{\beta_i}v_{ij} = (-1)^{k+1}\sum_{\ell=1}^{k+2} w_\ell u_{i,j'+\ell}$$

$$= \frac{(-1)^{k+1}}{\|e_i\|_2}\sum_{\ell=1}^{k+2}(-1)^{j'+\ell+1}w_\ell e_{i,j'+\ell}$$

$$= \frac{(-1)^{j'+1}}{\|e_i\|_2}\sum_{\ell=1}^{k+2}|w_\ell|\big(M_{i,j'+\ell} + L_{i,j'+\ell} + R_{i,j'+\ell} + \delta_{i,j'+\ell}\big). \tag{A.19}$$

We first work on the terms in the above sum involving $M_{i,j'+\ell}$, $\ell \in [k+2]$. Note that for $t \in \mathbb{C}$,

$$\sum_{\ell=1}^{k+2} |w_\ell| t^{\ell-1} = (1+t)^{k+1} = t^{(k+1)/2} q(t), \quad \text{where } q(t) = (2+t+1/t)^{(k+1)/2}. \quad (A.20)$$

Therefore, recalling (A.11), we have

$$\sum_{\ell=1}^{k+2} |w_\ell| M_{i,j'+\ell} = \frac{z_{0i}^{\frac{N'-1}{2}-j'}}{|b_{+i}(z_{0i})|} \sum_{\ell=1}^{k+2} |w_\ell| z_{0i}^{-(\ell-1)} + (-1)^{N'-i} \frac{\bar{z}_{0i}^{\frac{N'-1}{2}-j'}}{|b_{+i}(\bar{z}_{0i})|} \sum_{\ell=1}^{k+2} |w_\ell| \bar{z}_{0i}^{-(\ell-1)}$$

$$= \frac{z_{0i}^{\frac{N'-1}{2}-j'}}{|b_{+i}(z_{0i})|} z_{0i}^{-(k+1)/2} q(z_{0i}) + (-1)^{N'-i} \frac{\bar{z}_{0i}^{\frac{N'-1}{2}-j'}}{|b_{+i}(\bar{z}_{0i})|} \bar{z}_{0i}^{-(k+1)/2} q(\bar{z}_{0i}), \quad (A.21)$$

where in the last line we have used the fact that $q(t) = q(1/t)$. Recall also that $z_{0i} = z_0(\tau_i)$, where $\tau_i$ denotes the $i$th eigenvalue of $T$, i.e., $\tau_i = 4^{k+1} - \beta_i$. By definition, $z_{0i}$ is a unit-modulus root of the characteristic polynomial

$$p_{\tau_i}(t) = 4^{k+1} - (2+t+1/t)^{k+1} - \tau_i = \beta_i - (2+t+1/t)^{k+1}, \quad (A.22)$$

and therefore it holds that $q(z_{0i}) = q(\bar{z}_{0i}) = \sqrt{\beta_i}$. Continuing on from (A.21), we have

$$\sum_{\ell=1}^{k+2} |w_\ell| M_{i,j'+\ell} = \sqrt{\beta_i} \left( z_{0i}^{-(k+1)/2} + \bar{z}_{0i}^{-(k+1)/2} \right) M_{i,j'-1}. \quad (A.23)$$

Similar logic holds for the terms in (A.19) involving $L_{i,j'+\ell}$, $R_{i,j'+\ell}$, $\ell \in [k+2]$. First, we reexpress the definition in (A.12) as

$$L_{ij} = \sum_{\kappa=1}^{k} L_{i\kappa} z_\kappa(\tau_i)^{-j}.$$

Then, again applying (A.20), we have

$$\sum_{\ell=1}^{k+2} |w_\ell| L_{i,j'+\ell} = \sum_{\kappa=1}^{k} L_{i\kappa} z_\kappa(\tau_i)^{-1} q(z_\kappa(\tau_i)).$$

For each $\kappa \in [k]$, recall that $z_\kappa(\tau_i)$ is a root of the characteristic polynomial in (A.22) with modulus larger than 1, and hence $q(z_\kappa(\tau_i)) = \pm\sqrt{\beta_i}$. From the last display, this means we can write

$$\sum_{\ell=1}^{k+2} |w_\ell| L_{i,j'+\ell} = \sqrt{\beta_i} \sum_{\kappa=1}^{k} \sigma_{i\kappa} L_{i\kappa} z_\kappa(\tau_i)^{-j'-1}, \quad (A.24)$$

for signs $\sigma_{i\kappa} \in \{-1, 1\}$, $\kappa \in [k]$. Based on its definition in (A.13), we also have

$$\sum_{\ell=1}^{k+2} |w_\ell| R_{i,j'+\ell} = \sqrt{\beta_i} \overline{\left( \sum_{\kappa=1}^{k} \sigma_{i\kappa} L_{i\kappa} z_\kappa(\tau_i)^{-(N'-j')} \right)}. \quad (A.25)$$

Putting together (A.19), (A.23), (A.24), (A.25), and canceling out the common factor of $\sqrt{\beta_i}$, we have

$$v_{ij} = \frac{(-1)^{j'+1}}{\|e_i\|_2} \Bigg[ \left( z_{0i}^{-(k+1)/2} + \bar{z}_{0i}^{-(k+1)/2} \right) M_{i,j'-1} + \sum_{\kappa=1}^{k} \sigma_{i\kappa} L_{i\kappa} z_\kappa(\tau_i)^{-j'-1} +$$

$$\overline{\left( \sum_{\kappa=1}^{k} \sigma_{i\kappa} L_{i\kappa} z_\kappa(\tau_i)^{-(N'-j')} \right)} + \frac{\delta_{ij}}{\sqrt{\beta_i}} \Bigg].$$

Thus, using the fact that $|z_{0i}| = 1$ and $|z_\kappa(\tau_i)| > 1$, $\kappa \in [k]$,

$$|v_{ij}| \leq \frac{2}{\|e_i\|_2} \left( |M_{i,j'-1}| + \sum_{\kappa=1}^{k} |L_{i\kappa}| + \frac{|\delta_{ij}|}{\sqrt{\beta_i}} \right).$$

It can be shown from the form of the positive Wiener-Hopf factor $b_+(t,s)$ in (A.9) that $L_{i\kappa} = O(1)$, $\kappa \in [k]$, uniformly in $i$. Furthermore, as already shown in the proof of Lemma A.3, we know that $|M_{ij}|/\|e_i\|_2 = O(1/\sqrt{N'})$ uniformly over $i,j$, and also $\|e_i\|_2 = \Omega(\sqrt{N'})$, uniformly over $i$. Lastly, $|\delta_{ij}|/\sqrt{\beta_i} \leq (2/\pi)^{2k+2}|\delta_{ij}|N^{2k+2}$, where we have lower bounded the smallest singular value of $D$ using (A.8) and the inequality $\sin(x) \geq x/2$ for small enough $x$. This does not pose any problems, because the remainder term $\delta_{ij}$ is actually smaller than any polynomial in $N$, uniformly over $i,j$, according to Theorem 2.5 of Bogoya et al. [1]. Therefore, combining all of this with the last display, we have $|v_{ij}| = O(1/\sqrt{N'})$, uniformly over $i$ and interior coordinates $j$.

**Bounding the boundary elements.** Consider the "inverse" relationship to (A.18),

$$Dv_i = \sqrt{\beta_i}u_i, \quad i \in [N']. \tag{A.26}$$

Since $\beta_i \leq 4^{k+1}$, $i \in [N']$, and the vectors $u_i$, $i \in [N']$ are incoherent from Lemma A.3, we have

$$\|Dv_i\|_\infty \leq \frac{\mu}{\sqrt{N'}}, \quad i \in [N'],$$

for a constant $\mu > 0$ depending only on $k$, or more explicitly,

$$\left|\sum_{\ell=1}^{k+2} w_\ell v_{i,j+\ell-1}\right| \leq \frac{\mu}{\sqrt{N'}}, \quad i,j \in [N'].$$

Fix an arbitrary $i \in [N']$, and consider $j = k+1$. By the above display, the triangle inequality, and the observation that $|w_1| = 1$,

$$|v_{i,k+1}| \leq \frac{\mu}{\sqrt{N'}} + \sum_{\ell=2}^{k+2} |w_\ell||v_{i,k+\ell}| \leq \frac{c}{\sqrt{N'}},$$

for a constant $c > 0$ depending only on $k$, where in the second inequality we used the incoherence of the right singular vectors over the interior elements, as shown previously. Continuing on in the same manner verifies the incoherence property at all positions $j = k, \ldots, 1$, and similarly, at all positions $j = N'+1, \ldots, N$. This completes the proof. $\qquad\square$

**Lemma A.5.** *Abbreviate $D = D_{1d}^{(k+1)} \in \mathbb{R}^{(N-k-1)\times N}$, and use the notation $\mathrm{abs}(A)$ to denote the elementwise absolute value of a matrix $A$. Consider eigendecompositions*

$$DD^T = U\Lambda U^T, \quad \mathrm{abs}(DD^T) = U_+\Lambda U^T.$$

*Then:*

*(a)* $\Lambda = \Lambda_+$;

*(b)* $\mathrm{abs}(U) = \mathrm{abs}(U_+)$.

*Proof.* Denote $N' = N - k - 1$. Let $S \in \mathbb{R}^{N'\times N'}$ be the alternating sign diagonal matrix with diagonal elements $1, -1, 1, -1, \ldots$. Note that $S^{-1} = S^T = S$. From the relationship

$$DD^T = S^{-1}\mathrm{abs}(DD^T)S$$

we conclude that $DD^T$ and $\mathrm{abs}(DD^T)$ are similar, i.e., $\Lambda = \Lambda_+$. From their eigendecompositions,

$$U\Lambda U^T = SU_+\Lambda U_+^T S^T$$

we also see that $U = SU_+$ which implies $\mathrm{abs}(U) = \mathrm{abs}(U_+)$. $\qquad\square$

## A.8 Proof of Lemma 3

Denote

$$\widetilde{Z}_d = \left\{x = (x_1, \ldots, x_d) \in Z_d : x_j \leq 1 - (k+1)/N, j = 1, \ldots, d\right\}.$$

Pick an arbitrary $\theta \in \mathcal{H}_d^{k+1}(L)$, corresponding to discretizations of $f \in H(k+1, L; [0,1]^d)$. The bound (15) holds at any $x \in \widetilde{Z}_d$, and the fact that $\delta(N) \leq cL/N$ is verified by Lemma A.6. The KTF penalty is then

$$\|\widetilde{\Delta}^{(k+1)}\theta\|_1 = \sum_{x \in \widetilde{Z}_d} \left|\left(D_{x_j^{k+1}}\theta\right)(x)\right| \leq cnLN^{k-1} = cLn^{1-(k+1)/d},$$

recalling $N = n^{1/d}$.

## A.9 Lemma A.6

The following lemma follows standard calculations in numerical analysis, e.g., as in Strikwerda [7].

**Lemma A.6.** *Let $f \in H(k+1, L; [0,1]^d)$. The $k$th order forward discrete difference along a unit direction $v \in \mathbb{R}^d$, with step size $h > 0$, obeys at any point $x \in [0,1]^d$,*

$$\left| \frac{1}{h^k} (D_{v^k} \theta)(x) - \frac{\partial^k}{\partial v^k} f(x) \right| \leq cLh,$$

*where $c > 0$ is a constant depending only on $k$, provided that $x + khv \in [0,1]^d$ (so that the discrete approximation is well-defined).*

*Proof.* By Taylor expanding $f$ around $x$ at $x, x+hv, x+2hv, \ldots, x+khv$, we have

$$f(x) = f(x),$$

$$f(x+hv) = f(x) + \frac{\partial}{\partial v} f(x) h + \frac{1}{2} \frac{\partial^2}{\partial v^2} f(x) h^2 + \ldots + \frac{1}{k!} \frac{\partial^k}{\partial v^k} f(x) h^k + r(h),$$

$$f(x+2hv) = f(x) + \frac{\partial}{\partial v} f(x)(2h) + \frac{1}{2} \frac{\partial^2}{\partial v^2} f(x)(2h)^2 + \ldots + \frac{1}{k!} \frac{\partial^k}{\partial v^k} f(x)(2h)^k + r(2h),$$

$$\vdots$$

$$f(x+khv) = f(x) + \frac{\partial}{\partial v} f(x)(kh) + \frac{1}{2} \frac{\partial^2}{\partial v^2} f(x)(kh)^2 + \ldots + \frac{1}{k!} \frac{\partial^k}{\partial v^k} f(x)(kh)^k + r(kh),$$

where $r(ih)$ is integral form of the remainder in the expansion for $x + ihv$, satisfying

$$|r(ih)| = \left| \frac{1}{k!} \int_0^{ih} \frac{\partial^{k+1}}{\partial v^{k+1}} f(x+tv) t^k \, dt \right| \leq \frac{k^{k+1}}{(k+1)!} Lh^{k+1}, \quad i = 1, \ldots, k.$$

(Note that such integrals are well-defined since Lipschitz continuity of $\partial^k f / \partial v^k$ implies that the $(k+1)$st derivative $\partial^{k+1} f / \partial v^{k+1}$ exists almost everywhere and is Lebesgue integrable, by Rademacher's theorem.) In the inequality above, we invoked the Holder property, recalling $f \in H(k+1, L; [0,1]^d)$.

Now denote the $k$th order forward difference coefficients by

$$w_i = (-1)^{k+i-1} \binom{k}{i-1}, \quad i = 1, \ldots, k+1.$$

Inverting the above $(k+1) \times (k+1)$ system of equations (from the $k+1$ Taylor expansions), and inspecting the last equality in the inverted system, gives

$$\frac{\partial^k}{\partial v^k} f(x) h^k = \sum_{i=1}^{k+1} w_i \Big( f(x + (i-1)hv) - r((i-1)h) \Big) = (D_{v^k} \theta)(x) - \sum_{i=1}^{k+1} w_i r((i-1)h).$$

Using our previous bound on the magnitude of remainders, we see

$$\left| (D_{v^k} \theta)(x) - \frac{\partial^k}{\partial v^k} f(x) h^k \right| \leq \frac{k^{k+1}}{(k+1)!} \sum_{i=1}^{k+1} |w_i| Lh^{k+1},$$

and dividing through by $h^k$ gives the claimed result. $\square$

## A.10 Proof of Lemma 4

We need only to construct a single counterexample for each $k, d \geq 1$. We give such a construction for $d = 2$ and $k = 1$; all other cases follows similarly. Consider a function $f : [0,1]^d \to \mathbb{R}$ defined by $f(x) = Mx_1 + x_2$, and let $\theta \in \mathbb{R}^n$ contain the evaluations of $f$ over the grid $Z_2$. As $f$ is linear, it is clearly an element of $H(2, 1; [0,1]^2)$. But, for any $x$ on the left boundary of $Z_2$,

$$\|\Delta^{(2)} \theta\|_1 \geq \left| f\left(x + \frac{e_1}{N}\right) + f\left(x - \frac{e_2}{N}\right) + f\left(x + \frac{e_2}{N}\right) - 3f(x) \right| = \left| f\left(x + \frac{e_1}{N}\right) - f(x) \right| = Mn^{1/2},$$

Since $M$ can be arbitrary, this proves the result.

## A.11  Proof of Theorem 4

We will show that

$$R\big(\mathcal{H}_d^{k+1}(L_n)\big) = \Omega(n^{-\frac{2k+2}{2k+2+d}} L_n^{\frac{2d}{2k+2+d}}). \tag{A.27}$$

Taking $L_n = C_n/n^{1-(k+1)/d}$ and applying Lemma 3 would then establish the result.

The result is "nearly" a textbook result on Holder classes in nonparametric regression. A standard result (e.g., see Chapter 2.8 of Korostelev and Tsybakov [4]) is that, in a model

$$y_i = f_0(x_i) + \epsilon_i, \quad \epsilon_i \overset{\text{i.i.d.}}{\sim} N(0, \sigma^2), \; i = 1, \dots, n$$

where the design points $x_i \in [0,1]^d$, $i = 1, \dots, n$ are fixed and arbitrary, we have

$$\inf_{\hat{f}} \sup_{f_0 \in H(k+1, L_n; [0,1]^d)} \mathbb{E}\|\hat{f} - f_0\|_2^2 = \Omega(n^{-\frac{2k+2}{2k+2+d}} L_n^{\frac{2d}{2k+2+d}}), \tag{A.28}$$

where $\|\cdot\|_2$ denotes the $L_2$ norm on functions, defined as

$$\|f\|_2^2 = \int_{[0,1]^d} f(x)^2 \, dx.$$

Note that we can rewrite the desired result (A.27) as

$$\inf_{\hat{f}} \sup_{f_0 \in H(k+1, L_n; [0,1]^d)} \mathbb{E}\|\hat{f} - f_0\|_n^2 = \Omega(n^{-\frac{2k+2}{2k+2+d}} L_n^{\frac{2d}{2k+2+d}}), \tag{A.29}$$

where the design points are $\{x_1, \dots, x_n\} = Z_d$, the regular lattice on $[0,1]^d$, and where $\|\cdot\|_n$ denotes the empirical norm on functions, defined as

$$\|f\|_n^2 = \frac{1}{n} \sum_{i=1}^n f(x_i)^2.$$

The proof of (A.28) reduces the estimation problem to a multiple hypothesis testing problem, and then constructs a sufficiently hard set of hypothesis by taking linear combinations of kernel "bump" functions and applying the Varshamov–Gilbert lemma (e.g., see Sections 2.7, 2.8 of Korostelev and Tsybakov [4], or Section 2.6 of Tsybakov [8]). But in the standard construction, the bump functions are not only orthogonal with respect to the $L_2$ inner product, but also with respect to the empirical inner product, since their supports are nonoverlapping. Thus the exact same sequence of arguments leads to (A.29), i.e., leads to (A.27), provided the empirical norm a bump function is at least of the same order as its $L_2$ norm, as verified below.

Consider a partition of $[0,1]^d$ into $m \asymp n^{d/(2k+2+d)}$ hypercubes, each hypercube having side length $h = 1/m^{1/d} \asymp n^{-1/(2k+2+d)}$. Denote by $z_i$, $i = 1, \dots, m$ the hypercube centers and consider bump functions $\varphi_i(x) = \varphi(x - z_i)$, $i = 1, \dots, n$, where

$$\varphi(x) = h^{k+1} K\left(\frac{2\|x\|_2}{h}\right), \quad \text{where } K(u) = \exp\left(\frac{-1}{1 - u^2}\right) 1\{|u| < 1\}.$$

In the $L_2$ norm, it holds that $\|\varphi_i\|_2^2 \asymp h^{2k+2+d}$, $i = 1, \dots, n$. We want to show the empirical norms are lower bounded at the same rate. By symmetry, it suffices to study one bump function, say, $\varphi_1$. Denote by $U_1$ the set of grid points lying in a sphere of radius $h/(2\sqrt{2})$ around $z_1$. As $K(u) \geq 1/e^2$ for $|u| \leq 1/\sqrt{2}$, we have $\varphi_1(x) \geq h^{k+1}/e^2$ for $x \in U_1$. But the number of elements in $U_1$ is on the order of $nh^d$, and therefore $\|\varphi_1\|_n^2 = \Omega(h^d h^{2k+2}) = \Omega(h^{2k+2+d})$, as desired. $\qquad\square$

## A.12  Proof of Theorem 5

Define a class

$$\mathcal{S}_d^{k+1} = \{\theta \in \mathbb{R}^n : \|\Delta^{(k+1)}\theta\|_2 \leq B_n\} = \{\theta \in \mathbb{R}^n : \theta^T L^{k+1}\theta \leq B_n^2\}.$$

Notice that $\mathcal{S}_d^{k+1}(B_n) \subseteq \mathcal{T}_d^{k+1}(C_n)$ provided $B_n = C_n/\sqrt{r}$, where $r \asymp n$ is the number of rows of of $\Delta^{(k+1)}$, owing to the simple inequality $\|x\|_1 \leq \sqrt{r}\|x\|_2$ for $x \in \mathbb{R}^n$. We will show that

$$R\big(\mathcal{S}_d^{k+1}(B_n)\big) = \Omega(n^{-\frac{d}{2k+2+d}} B_n^{\frac{2d}{2k+2+d}}). \tag{A.30}$$

Taking $B_n \asymp C_n/\sqrt{n}$ would then give the result.

Letting $L = U\Lambda U^T$ be an eigendecomposition, and note that for any estimator $\hat{\theta}$ of $\theta_0$,

$$\|\hat{\theta} - \theta_0\|_2 = \|U^T\hat{\theta} - U^T\theta_0\|_2,$$

which means that we may rotate the parameter space and equivalently consider the minimax error over the rotated class

$$\widetilde{\mathcal{S}}_d^{k+1} = \left\{ \gamma \in \mathbb{R}^n : \sum_{i=1}^{n} \lambda_i^{k+1}\gamma_i^2 \le B_n^2 \right\},$$

where we have denoted the eigenvalues (diagonal elements of $\Lambda$) as $\lambda_i$, $i \in [n]$. We will now seek to embed a hyperrectangle in the above class and make use of results of Donoho et al. [3].

Write $\gamma = (\alpha, \beta) \in \mathbb{R} \times \mathbb{R}^{n-1}$, and order $\lambda_1 \le \lambda_2 \le \ldots \le \lambda_n$, so the above class becomes

$$\widetilde{\mathcal{S}}_d^{k+1} = \left\{ (\alpha, \beta) \in \mathbb{R} \times \mathbb{R}^{n-1} : \sum_{i=2}^{n} \lambda_i^{k+1}\beta_i^2 \le B_n^2 \right\} := \mathbb{R} \times \mathcal{E}(B_n),$$

where we have used the fact that $\lambda_1 = 0$. (Here and henceforth, although unconventional, we will index $\beta$ according to components $i = 2, \ldots, n$, rather than $i = 1, \ldots, n-1$, because it simplifies notation later.) The minimax risk (writing $\gamma_0 = U^T\theta_0$, and $\gamma_0 = (\alpha_0, \beta_0)$) satisfies

$$\inf_{\hat{\gamma}} \sup_{\gamma_0 \in \widetilde{\mathcal{S}}_d^{k+1}} \frac{1}{n}\mathbb{E}\|\hat{\gamma} - \gamma_0\|_2^2 = \frac{\sigma^2}{n} + \inf_{\hat{\beta}} \sup_{\beta_0 \in \mathcal{E}(B_n)} \frac{1}{n}\mathbb{E}\|\hat{\beta} - \beta_0\|_2^2.$$

We focus on the second term. The ellipsoid $\mathcal{E}(B_n)$ is compact, convex, orthosymmetric and quadratically convex, the latter property as defined in Donoho et al. [3]. We can therefore use Lemma 6 and Theorem 7 in their work to conclude that the minimax risk over $\mathcal{E}(B_n)$ is at least four-fifths of the minimax linear risk of its hardest hyperrectangle,

$$\inf_{\hat{\beta}} \sup_{\beta_0 \in \mathcal{E}(B_n)} \frac{1}{n}\mathbb{E}\|\hat{\beta} - \beta_0\|_2^2 \ge \frac{4}{5} \sup_{H \subseteq \mathcal{E}(B_n)} \inf_{\hat{\beta} \text{ linear}} \sup_{\beta_0 \in H} \frac{1}{n}\mathbb{E}\|\hat{\beta} - \beta_0\|_2^2, \qquad \text{(A.31)}$$

where the outer sup on the right-hand side is over hyperrectangles $H$ contained in $\mathcal{E}(B_n)$. Consider hyperrectangles parametrized by a threshold $\tau$,

$$H(\tau) = \{\beta \in \mathbb{R}^{n-1} : |\beta_i| \le t_i(\tau), \ i = 2, \ldots, n\},$$

where for all $i = 2, \ldots, n$, using multi-index notation $i = (i_1, \ldots, i_d)$, we let

$$t_{i+1}(\tau) = \begin{cases} B_n/(\sum_{i_1,\ldots,i_d \le \tau} \lambda_i^{k+1})^{1/2} & \text{if } i_1, \ldots, i_d \le \tau \\ 0 & \text{else.} \end{cases}$$

It is not hard to check that $H(\tau) \subseteq \mathcal{E}(B_n)$. The minimax linear risk over $H(\tau)$ decomposes, and can be evaluated exactly, as in Donoho et al. [3],

$$\inf_{\hat{\beta} \text{ linear}} \sup_{\beta_0 \in H(\tau)} \frac{1}{n}\mathbb{E}\|\hat{\beta} - \beta_0\|_2^2 = \frac{1}{n}\sum_{i=2}^{n} \frac{t_i(\tau)^2\sigma^2}{t_i(\tau)^2 + \sigma^2} = \frac{1}{n}\frac{(\tau^d - 1)\sigma^2 B_n^2}{B_n^2 + \sum_{i_1,\ldots,i_d \le \tau} \lambda_i^{k+1}}.$$

Lemma A.7 provides an upper bound on the sum in the denominator above, and plugging this in, we get

$$\inf_{\hat{\beta} \text{ linear}} \sup_{\beta_0 \in H(\tau)} \frac{1}{n}\mathbb{E}\|\hat{\beta} - \beta_0\|_2^2 \ge \frac{1}{n}\frac{(\tau^d - 1)\sigma^2 B_n^2}{B_n^2 + c\frac{\tau^{2k+2+d}}{N^{2k+2}}},$$

for a constant $c > 0$. This lower bound is maximized at $\tau \asymp (B_n^2 N^{2k+2})^{\frac{1}{2k+2+d}}$, in which case, we see

$$\inf_{\hat{\beta} \text{ linear}} \sup_{\beta_0 \in H(\tau)} \frac{1}{n}\mathbb{E}\|\hat{\beta} - \beta_0\|_2^2 = \Omega(n^{-\frac{d}{2k+2+d}} B_n^{\frac{2d}{2k+2+d}}).$$

Recalling (A.31), we have hence shown (A.30), and this completes the proof.

## A.13  Lemma A.7

This result slightly generalizes Lemma A.3 of Sadhanala et al. [6].

**Lemma A.7.** *Let $L \in \mathbb{R}^{n \times n}$ denote the Laplacian matrix of the $d$-dimensional grid graph with equal side lengths $N = n^{1/d}$, and let*

$$\lambda_{i_1,\ldots,i_d} = 4 \sum_{j=1}^{d} \sin^2 \left( \frac{\pi(i_j - 1)}{2N} \right), \quad i_1,\ldots,i_d \in [N]$$

*denote its eigenvalues. Then for any integer $k \geq 0$ and $\tau \in [N]$,*

$$\sum_{i_1,\ldots,i_d \leq \tau} \lambda_{i_1,\ldots,i_d}^{k+1} \leq c \frac{\tau^{2k+2+d}}{N^{2k+2}},$$

*for a constant $c > 0$ depending only on $k$ and $d$.*

*Proof.* The proof follows the same chain of arguments as that for Lemma A.3 in Sadhanala et al. [6]. Using the fact that $\sin(x) \leq x$ for all $x \geq 0$,

$$\sum_{i_1,\ldots,i_d \leq \tau} \lambda_{i_1,\ldots,i_d}^{k+1} \leq \frac{\pi^{2k+2}}{4^k N^{2k+2}} \sum_{i_1,\ldots,i_d \leq \tau} \left( (i_1 - 1)^2 + \ldots + (i_d - 1)^2 \right)^{k+1}$$

$$\leq \frac{\pi^{2k+2}}{4^k N^{2k+2}} \tau^{d-1} \sum_{i=1}^{\tau} (i-1)^{2k+2}$$

$$\leq c \frac{\tau^{2k+2+d}}{N^{2k+2}}.$$

$\square$