[Reviews · NeurIPS 2017]

Reviewer 1



This paper considers graph-structured signal denoising problem. The particular structure enforced involves a total variation type penalty involving higher order discrete derivatives. Optimal rates for penalized least-squares estimator is established and minimax lower bound is established. For the grid filtering case the upper bounds are established under an assumed conjecture. The paper is well written. The main proof techniques of the paper are borrowed from previous work but there are some additional technicalities required to establish the current results. I went over the proofs (not 100% though) and it seems there is no significant issues with it. Overall I find the results of the paper as interesting contributions to trend filtering literature and I recommend to accept the paper. Minor: it is not clear why lines 86-87 are there in the supplementary material.

Reviewer 2



This paper studies the problem of estimating function values over a d-dimensional grid. The authors propose a new grid trend filtering penalty, they show estimation rates of their newly proposed algorithm as well as they improve previous rates for graph-trend filtering for the special case d = 2. They complement these rates by showing lower bounds which nearly match the estimation rates of the grid and graph trend filtering. The authors also comment on how TV-graph and TV-grid classes relate to classical Holder classes. The authors should double check the paper for typos as I saw several typos: 159: possibleto 225: linear smoothers such that How did the authors choose the tuning parameters in practice to produce Figure 1? What is the intuition why Conjecture 1 holds? As it is written it seems the authors assume it just to make their proof go through.

Reviewer 3



This paper established the lower and upper bounds of two total variation (TV) classes: TV-graph and TV-grid under the generalized lasso estimator with certain chosen penalty operators, and proved that the two corresponding trend filtering methods are minimax optimal. The motivation of broaden the existing TV denoising results to quantify the minimax rate in higher order graph/grids is clear, the presentation of the paper is nice, and the math derivation is clear. The content is mathematically/statistically solid, thought not clear about the influence on imaging due to reviewer's background. (BTW, has the incoherence of A.5 in supplemental material figured out already?)